# Enabling X-ray free electron laser crystallography for challenging biological systems from a limited number of crystals

Monarin Uervirojnangkoorn[1], Oliver B Zeldin[1], Artem Y Lyubimov[1], Johan Hattne[2], Aaron S Brewster[3], Nicholas K Sauter[3], Axel T Brunger[1,4,5]*, William I Weis[1,5,6]*

[1]Department of Molecular and Cellular Physiology, Stanford University, Stanford, United States; [2]Janelia Research Campus, Ashburn, United States; [3]Physical Biosciences Division, Lawrence Berkeley National Laboratory, Berkeley, United States; [4]Department of Neurology and Neurological Sciences, Howard Hughes Medical Institute, Stanford University, Stanford, United States; [5]Department of Photon Science, Stanford University, Stanford, United States; [6]Department of Structural Biology, Stanford University, Stanford, United States

**Abstract** There is considerable potential for X-ray free electron lasers (XFELs) to enable determination of macromolecular crystal structures that are difficult to solve using current synchrotron sources. Prior XFEL studies often involved the collection of thousands to millions of diffraction images, in part due to limitations of data processing methods. We implemented a data processing system based on classical post-refinement techniques, adapted to specific properties of XFEL diffraction data. When applied to XFEL data from three different proteins collected using various sample delivery systems and XFEL beam parameters, our method improved the quality of the diffraction data as well as the resulting refined atomic models and electron density maps. Moreover, the number of observations for a reflection necessary to assemble an accurate data set could be reduced to a few observations. These developments will help expand the applicability of XFEL crystallography to challenging biological systems, including cases where sample is limited.

*For correspondence: brunger@ stanford.edu (ATB); bill.weis@ stanford.edu (WIW)

## Introduction

Radiation damage often limits the resolution and accuracy of macromolecular crystal structures (*Garman, 2010*; *Zeldin et al., 2013*). Femtosecond X-ray free electron laser (XFEL) pulses enable the possibility of visualizing molecular structures before the onset of radiation damage, and allow the dynamics of chemical processes to be captured (*Solem, 1986*; *Neutze et al., 2000*). Thus, from the first XFEL operation at the Linac Coherent Light Source (LCLS) in 2009, there has been considerable effort dedicated to the development of methods to utilize this rapid succession of bright pulses for macromolecular crystallography, with the aim of obtaining damage-free, chemically accurate structures. Most of the structures reported from XFELs to date use a liquid jet to inject small crystals into the beam (*DePonte et al., 2008*; *Sierra et al., 2012*; *Weierstall et al., 2014*), but diffraction data have also been measured from crystals placed in the beam with a standard goniometer setup (*Cohen et al., 2014*; *Hirata et al., 2014*). In both cases, the illuminated volume diffracts before suffering damage by a single XFEL pulse. Because the crystal is effectively stationary during the 10–50 fs exposure, 'still' diffraction patterns are obtained, in contrast to standard diffraction data collection where the sample is rotated through a small angle during the exposure.

**eLife digest** Large biological molecules (or macromolecules) have intricate three-dimensional structures. X-ray crystallography is a technique that is commonly used to determine these structures and involves directing a beam of X-rays at a crystal that was grown from the macromolecule of interest. The macromolecules in the crystal scatter the X-rays to produce a diffraction pattern, and the crystal is rotated to provide further diffraction images. It is then possible to work backwards from these images and elucidate the structure of the macromolecule in three dimensions.

X-ray beams are powerful enough to damage crystals, and scientists are developing new approaches to overcome this problem. One recent development uses 'X-ray free electron lasers' to circumvent the damage caused to crystals. However, early applications of this approach required many crystals and thousands to millions of diffraction patterns to be collected—largely because methods to process the diffraction data were far from optimal.

Uervirojnangkoorn et al. have now developed a new data-processing procedure that is specifically designed for diffraction data obtained using X-ray free electron lasers. This method was applied to diffraction data collected from crystals of three different macromolecules (which in this case were three different proteins). For all three, the new method required many fewer diffraction images to determine the structure, and in one case revealed more details about the structure than the existing methods.

This new method is now expected to allow a wider range of macromolecules to be studied using crystallography with X-ray free electron lasers, including cases where very few crystals are available.

Extracting accurate Bragg peak intensities from XFEL diffraction data is a substantial challenge. An XFEL data set comprises 'still' diffraction patterns generally containing only partially recorded reflections, typically from randomly oriented crystals. The full intensity then has to be estimated from the observed partial intensity observations. Most XFEL diffraction data processing approaches reported to date have approximated the full intensity by the so-called "Monte Carlo" method, in which thousands of partial intensity observations of a given reflection are summed and normalized by the number of observations, which assumes that these observations sample the full 3D Bragg volume. Because a single diffraction image—in which each observed reflection samples only part of each reflection intensity–contains much less information than a small continuous wedge of diffraction data (as used in conventional crystallography), this method requires a very large number of crystals to ensure convergence of the averaged partial reflection intensities to the full intensity value (*Kirian et al., 2010*). Moreover, shot-to-shot differences in pulse intensity and energy spectrum that arise from the self-amplified stimulated emission (SASE) process (*Kondratenko and Saldin, 1979*; *Bonifacio et al., 1984*), along with differences in illuminated crystal volume, mosaicity, and unit-cell dimensions, contribute to intensity variation of the equivalent reflections observed on different images. These differences are assumed to be averaged out by the Monte Carlo method (*Hattne et al., 2014*). Thus, accurate determination of these parameters for each diffraction image should, in principle, provide more accurate integrated intensities, and converge with fewer measurements. Furthermore, it is desirable to assemble a data set from as few diffraction images as possible, since the potential of XFELs has been limited by the very large amounts of sample required for the Monte Carlo method, compounded by severe limitations in the availability of beamtime.

In the 1970's, the Harrison and Rossmann groups developed 'post-refinement' methods (*Rossmann et al., 1979*; *Winkler et al., 1979*), in which the parameters that determine the location and volume of the Bragg peaks are 'post'-refined against a reference set of fully recorded reflections following initial indexing and integration of rotation data. Accurate estimation of these parameters, including the unit-cell lengths and angles, crystal orientation, mosaic spread, and beam divergence enables accurate calculation of what fraction of the reflection intensity was recorded on the image, i.e., its 'partiality', which is then used to correct the measurement to its fully recorded equivalent. Applied to virus crystals, for which only a few images can typically be collected before radiation damage becomes significant, post-refinement made it possible to obtain high-quality diffraction data sets collected from many crystals (*Rossmann et al., 1979*; *Winkler et al., 1979*).

The implementation of post-refinement for XFEL diffraction data poses unique challenges. Firstly, since XFEL diffraction data generally do not contain fully recorded reflections, the initial scaling and merging of images is difficult. Secondly, since the XFEL diffraction images are stills rather than rotation data, different approaches are required for the correction of measurements to determine the full spot equivalent. Other schemes for implementing post-refinement of XFEL diffraction data have been described previously, but thus far they have been only applied to simulated XFEL data (*White, 2014*), and to pseudo-still images collected using monochromatic synchrotron radiation (*Kabsch, 2014*).

We have developed a new post-refinement procedure specifically designed for diffraction data from still images collected from crystals in random orientations. We implemented our method in a new computer program, *prime* (**p**ost-**r**ef**i**nement and **me**rging), that post-refines the parameters needed for calculating the partiality of reflections recorded on each still image. We describe here our method and demonstrate that post-refinement greatly improves the quality of the diffraction data from XFEL diffraction experiments with crystals of three different proteins. We show that our post-refinement procedure allows complete data sets to be extracted from a much smaller number of diffraction images than that necessary when using the Monte Carlo method. Thus, this development will help make XFEL crystallography accessible to many challenging problems in biology, including those for which sample quantity is a major limiting factor.

## Results

### Notation

Units are arbitrary unless specified in parenthesis.

$I_{obs}$, observed intensity.

$I_{ref}$, reference intensity.

$w$, weighting term (inverse variance of the observed intensity).

$G$, function of linear scale ($G_0$) and resolution-dependent ($B$) factors that scales the different diffraction images to the reference set.

$Eoc$, Ewald-offset correction function.

$r_h$, offset reciprocal-space distance from the center of the reflection to the Ewald sphere ($\mathring{A}^{-1}$).

$r_p$, radius of the disc of intersection between the reciprocal lattice point and the Ewald sphere ($\mathring{A}^{-1}$).

$r_s$, radius of the reciprocal lattice point ($\mathring{A}^{-1}$).

$\theta_x$, $\theta_y$, $\theta_z$, crystal rotation angles (see *Figure 1A*; °).

$\gamma_0$, parameter for *Equation 3* ($\mathring{A}^{-1}$).

$\gamma_e$, energy spread and unit-cell variation (see *Equation 3*; $\mathring{A}^{-1}$).

$\gamma_x$ and $\gamma_y$, beam divergence (see *Equation 4*; $\mathring{A}^{-1}$).

$\{uc\}$, unit-cell dimensions ($a$, $b$, $c$ ($\mathring{A}$), $\alpha$, $\beta$, and $\gamma$ (°)).

$V_c$, reciprocal-lattice volume correction function ($\mathring{A}^{-3}$).

$x_{obs}$ and $x_{calc}$, observed and predicted spot positions on the detector (mm).

$x$, position of the reciprocal lattice point ($\mathring{A}^{-1}$).

$S$, displacement vector from the center of the Ewald sphere to $x$ ($\mathring{A}^{-1}$).

$S_0$, incident beam vector with length 1/wavelength ($\mathring{A}^{-1}$).

$O$, orthogonalization matrix.

$R$, rotation matrix.

$f_L$ and $f_{LN}$, Lorentzian function and its normalized counterpart.

$\Gamma$, full width at half maximum (FWHM) of the Lorentzian function.

### Post-refinement overview

Partiality can be modeled by describing the full reflection as a sphere (*Figure 1A*). In a still diffraction pattern, assuming a monochromatic photon source, the observed intensity $I_{obs,h}$ for Miller index **h** is a thin slice through a three-dimensional reflection. To calculate partiality, we assume that the measurement is an areal (i.e., infinitely thin) sample of the volume (*Figure 1B*). The maximum partial intensity that can be recorded for a given reflection will occur when its center lies exactly on the Ewald sphere. By definition, the center of the reflection will be offset from the Ewald sphere by $r_h$, and the corresponding disc will have a radius $r_p$. The offset $r_h$ is determined by various experimental parameters, including the crystal orientation, unit-cell dimensions, and X-ray photon energy.

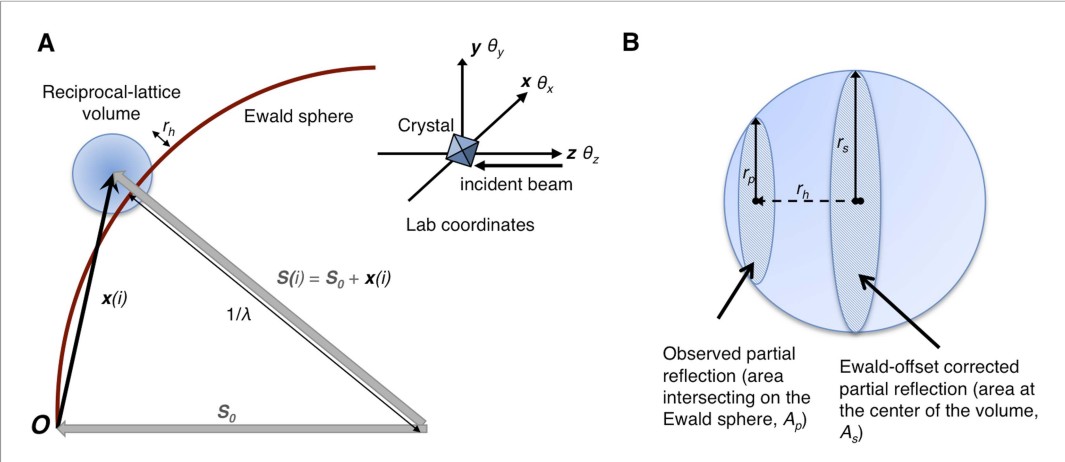

**Figure 1**. Geometry of the diffraction experiment and calculation of the Ewald-offset distance, $r_h$. (**A**) A reciprocal lattice point intersects the Ewald sphere. The inset shows the coordinate system used in *cctbx.xfel* and *prime*. The vector $\mathbf{S}_0$ represents the direction of the incident beam (–$z$-axis) and forms the radius of the Ewald sphere of length $1/\lambda$. The reciprocal lattice point $i$ is expressed in reciprocal lab coordinates using *Equation 5* as represented by the vector $\mathbf{x}_i$. The Ewald-offset distance, $r_h$, is the difference between the distance from the Ewald-sphere center to the reciprocal lattice point (length of $\mathbf{S}_i$) and $1/\lambda$. The inset shows the definition of the crystal rotation axes; they are applied in the following order: $\theta_z$, $\theta_y$, $\theta_x$. (**B**) Shown is the volume of a reciprocal lattice point with radius $r_s$. The offset $r_h$ defines the Ewald-offset correction $Eoc_{area}$, which is the ratio between the area intersecting the Ewald sphere, $A_p$, and the area at the center of the volume, $A_s$.

The offset distance is used to calculate the Ewald offset correction, $Eoc_{area}$, defined as the ratio between the areas defined by $r_p$ and $r_s$ (implemented as a smoothed correction function $Eoc_h$ as defined in 'Materials and methods'). The Ewald-offset corrected intensity is then converted to the full intensity in 3D by applying a volume correction factor, $V_c$.

We define the target $T_{pr}$ for the post-refinement of a partiality and scaling model by:

$$T_{pr} = \sum_h w_h \left( I_{obs,h} - G(G_0, B) \mathbf{Eoc}_h \left( \theta_x, \ \theta_y, \ \gamma_0, \gamma_e, \gamma_x, \gamma_y, \{uc\} \right) V_{c,h}^{-1} \times I_{ref,h} \right)^2 , \tag{1}$$

which minimizes the difference between the observed reflections $I_{obs}$ and a scaled and Ewald-offset corrected full intensity 'reference set' $I_{ref}$ using a least-squares method. The sum is over all observed reflections with Miller indices $\mathbf{h}$.

In alternate refinement cycles, we also minimize the deviations between predicted ($\mathbf{x}_{calc}$) and observed ($\mathbf{x}_{obs}$) spot positions on the detector using a subset of strong spots as has been suggested previously (*Hattne et al., 2014*; *Kabsch, 2014*):

$$T_{xy} = \sum_h \left( \mathbf{x}_{obs} - \mathbf{x}_{calc} \right)^2 . \tag{2}$$

Sets of parameters associated with each diffraction image, i.e., $G_0$, $B$, $\theta_x$, $\theta_y$, $\gamma_0$, $\gamma_e$, $\gamma_x$, $\gamma_y$ and the unit-cell constants, are iteratively refined in a series of 'microcycles' against the current reference set (*Figure 2*).

Procedures for generating the initial reference set $I_{ref}$(*initial*) are described below. After convergence of the microcycles, scaled full intensities are calculated from the observed partial intensities $I_{obs}$ by multiplication of the inverse of the Ewald-offset correction and the scale factor $G$, along with the volume correction factor $V_c$. These scaled full reflections are then merged for each unique Miller index, taking into account estimated errors of the observed intensities, $\sigma(I_{obs})$, and propagation of error estimates for the refined parameters. This merged and scaled set of full reflections is then used as the new reference set in the next round of post-refinement using the target functions (*Equations 1* and *2*, for details see 'Materials and methods'). These 'macrocycles' are repeated until convergence is achieved, after which the merged and scaled set of full intensities is provided to the user.

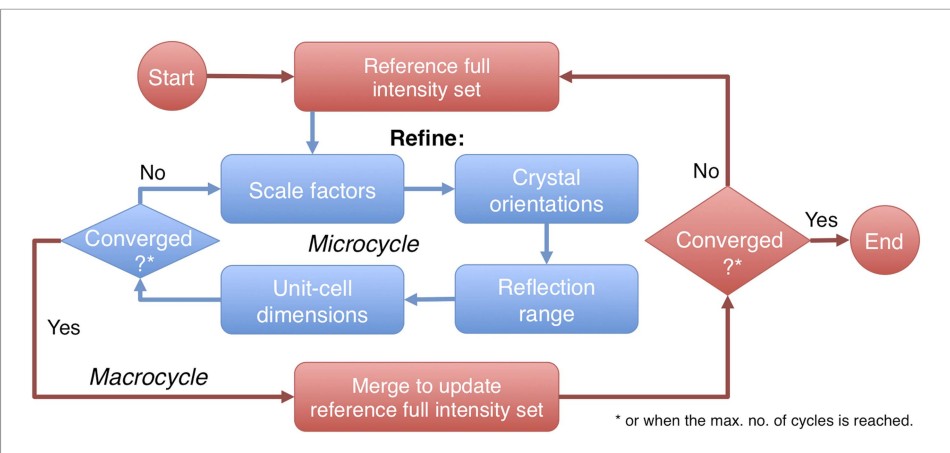

**Figure 2.** Post-refinement protocol. The flowchart illustrates the iterative post-refinement protocol, broken up into 'microcycles' that refine groups of parameters iteratively (blue boxes), and 'macrocycles'. At the beginning of first macrocycle, a reference diffraction data set is generated. At the end of each macrocycle, the reference diffraction data set is updated. Both the micro- and macrocycles terminate either when the refinement converges or when a user-specified maximum number of cycles is reached.

The *prime* program controls post-refinement of specified parameters in a particular microcycle (*Figure 2*). One can refine all parameters together, or selectively refine groups of parameters iteratively, starting from (1) a linear scale factor and a *B*-factor, (2) crystal orientations, (3) crystal mosaicity, beam divergence, and spectral dispersion, and (4) unit-cell dimensions. Space-group-specific constraints are used to limit the number of free parameters for the unit-cell refinement. A particular microcycle is completed when the target functions converge or when a specified number of iterations is reached; the program then generates the new reference intensity set to replace the current reference set for the next macrocycle. Finally, the program exits and outputs the latest merged reflection set either when the macrocycles converge or when a user-specified maximum number of cycles has been reached.

## Preparation of the observed intensities

The starting point for our post-refinement method is a set of indexed and integrated partial intensities, along with their estimated errors, obtained from still images. For this study, diffraction data and their estimated errors were obtained from the *cctbx.xfel* package (*Sauter et al., 2013*; *Hattne et al., 2014*), although in principle integrated diffraction data from any other program can be used. Observed intensities on the diffraction image were classified as 'spots' by the program Spotfinder (*Zhang et al., 2006*), which identifies Bragg spots by considering connected pixels with area and signal height greater than user-defined thresholds. By trial and error, we accepted reflections larger than 25 pixels with individual-pixel intensity more than 5 σ over background for myoglobin and hydrogenase (collected on a Rayonix MX325HE detector with pixel size of 0.08 mm and beam diameter [FWHM] of 50 μm). For thermolysin (collected on a Cornell-SLAC pixel array detector with pixel size of 0.1 mm and beam size of 2.25 μm²), where reflections are generally smaller, these values were 1 pixel and 5 σ. A full list of parameters is available on the *cctbx.xfel* wiki (http://cci.lbl.gov/xfel). Separate resolution cutoffs for each image were applied by *cctbx.xfel*, at resolutions where the average $I/\sigma(I)$ fell below 0.5 (*Hattne et al., 2014*).

Prior to post-refinement, the experimentally observed partial intensities need to be corrected by a polarization factor. The primary XFEL beam at LCLS is strongly polarized in the horizontal plane, and we calculate the correction factor as a function of the Bragg angle ($\theta$) and the angle $\phi$ between the sample reflection and the laboratory horizontal planes (*Kahn et al., 1982*; see 'Materials and methods'). For a stationary crystal and a monochromatic beam, a Lorentz factor correction is not applicable; the spectral dispersion of the SASE beam ($\delta E/E \sim 3 \times 10^{-3}$ for the data sets studied here) is accounted for by the $\gamma_e$ term (see 'Materials and methods').

### Generating the initial reference set and initial parameters

An essential step to initiate post-refinement is the generation of the initial reference set $I_{ref}$ (initial). This reference set has to be estimated from the available unmerged and unscaled partial reflection intensities after application of the polarization correction. For the results presented here, linear scale factors for each diffraction image were chosen to make the mean intensities of each diffraction image equal. Since this procedure can be affected by outliers in the observed intensities, we select a subset of reflections with user-specified resolution range and signal-to-noise ratio ($I/\sigma(I)$) cutoffs. From this selection, we calculate the mean intensity on each diffraction image and then scale each image to make the mean intensity of all images equal. We correct the scaled observed reflections to their Ewald-offset corrected equivalents using the starting parameters, and then merge the observations, taking into account the experimental $\sigma(I_{obs})$, to generate the initial reference set.

The initial values for crystal orientation, unit-cell dimensions, crystal-to-detector distance, and spot position on the detector were obtained from the refinement of these parameters by *cctbx.xfel*. The photon energy was that provided by the LCLS endstation system and is not refined. Initial values for the parameters of the reflection width model are described in the 'Materials and methods' section.

### Definition and comparison of data processing schemes

In order to separately assess the effects of scaling, the Ewald offset correction (*Equation 1*), and post-refinement, we refer to three alternative schemes for processing the diffraction data sets: (1) 'Averaged merged', in which intensities were generated by averaging all observed partial intensities from equivalent reflections without Ewald-offset correction and scaling; (2) 'Mean-intensity partiality corrected', in which intensities were generated by scaling the reflections to the mean intensity and also applying the Ewald-offset correction determined from the initial parameters obtained from the indexing and integration program, followed by merging; and (3) 'Post-refined', in which intensities were from the final set of scaled and merged full reflections after the convergence of post-refinement. We note that although the 'averaged merged' process is similar to the original Monte Carlo method (*Kirian et al., 2010*), the integrated, unmerged partial intensities used in our tests were obtained from the program *cctbx.xfel* (*Hattne et al., 2014*), which also refines various parameters on an image-by-image basis (*Sauter et al., 2014*).

### Quality assessment of post-refined data

We tested our post-refinement method on experimental XFEL diffraction data sets from three different crystallized proteins of known structure: myoglobin, hydrogenase, and thermolysin (*Table 1*). For quality assessment, we performed molecular replacement (MR) with Phaser (*McCoy et al., 2007*) using models with selected parts of the known structures omitted, followed by atomic model refinement with *phenix.refine* (*Afonine et al., 2012*), and inspection of ($mF_o$-$DF_c$) omit maps. We further used three different metrics: $CC_{1/2}$, and the crystallographic $R_{work}$ and $R_{free}$ of the fully refined atomic model. We then compared changes in the three quality metrics between merged XFEL diffraction data sets after scaling, partiality correction, and post-refinement. We also investigated the effect of reducing the number of images used by randomly selecting a subset from the full set of diffraction images and repeating the entire post-refinement, merging, MR and refinement processes using this subset.

Diffraction data for both myoglobin and hydrogenase were collected from frozen crystals mounted on a standard goniometer setup (*Cohen et al., 2014*), whereas the thermolysin data were collected using an electrospun liquid jet to inject nanocystals into a vacuum chamber (*Sierra et al., 2012*; *Bogan, 2013*). The completeness of each data set was better than 90% at the limiting resolution used in our tests (*Tables 2, 3, 4*). Each diffraction data set involved a different number of images due the differing diffraction quality of the crystals.

### Myoglobin

For myoglobin, we used both an XFEL diffraction data set consisting of 757 diffraction images (*Table 1*) collected by the SSRL-SMB group using a goniometer-mounted fixed-target grid (*Cohen et al., 2014*), and a randomly selected subset of 100 diffraction images. The diffraction images were from crystals in random orientations, with a single still image collected from each crystal.

**Table 1**. XFEL diffraction data sets used in this study

| | Myoglobin | *Clostridium pasteurianum* hydrogenase | Thermolysin |
|---|---|---|---|
| Space group | P6 | P4$_2$2$_1$2 | P6$_1$22 |
| Resolution used (Å) | 20.0–1.35 | 45.0–1.60 | 50.0–2.10 |
| Unit cell dimensions (Å) | *a* = *b* = 90.8, *c* = 45.6 | *a* = *b* = 111.2, *c* = 103.8 | *a* = *b* = 92.7, *c* = 130.5 |
| No. of unique reflections | 46,555 | 85,273 | 19,995 |
| No. of images* indexed | 757 | 177 | 12,692 |
| No. of images with spots to resolution used | 307 | 75 | 1957 |
| Average no. of spots on an image (to resolution used) | 1628 | 3640 | 352 |
| Energy spectrum | SASE† | SASE† | SASE† |
| Detector | Rayonix MX325HE | Rayonix MX325HE | CSPAD‡ |
| Sample delivery method | fixed target | fixed target | Electrospun jet |

*This is the number of images indexed using *cctbx.xfel* program, and in the case of thermolysin it is the number of images indexed for one of the two wavelengths.

†SASE: self-amplified spontaneous emission.

‡CSPAD: Cornell-SLAC pixel array detector.

## Convergence of post-refinement

Convergence properties for our post-refinement method for myoglobin are shown in *Figures 3 and 4*, and a representative example of the first macrocycle for a selected diffraction image is provided in *Figure 3*. The order of the three microcycle post-refinement iterations was: scale factors (SF—*Equation 17*), crystal orientation (CO—*Equation 5*), reciprocal spot size (RR—*Equations 3* and *4*), and unit-cell dimensions (UC—*Equation 5*). The partiality model target function $T_{pr}$ (*Equation 1*) markedly decreased in the first microcycle and fully converged in the last microcycle. The spot position residual $T_{xy}$ (*Equation 2*), also decreased both during post-refinement of the crystal orientation and the unit-cell parameters.

*Figure 5* shows the results for five macrocycles for post-refinement using the subset of 100 randomly selected still images of the myoglobin XFEL diffraction data set. The partiality model target function $T_{pr}$ (*Equation 1*) continually decreased in the first three macrocycles. The average spot position residual $T_{xy}$ (*Equation 2*) decreased in the first cycle and converged in the next cycle. The quality metric $CC_{1/2}$ also converged within the first three macrocycles.

Inaccuracies in the starting parameters obtained from indexing and integration of still images may limit the radius of convergence and the accuracy of the post-refined parameters. The sources of such errors will be the subject of future improvement in indexing and integration in *cctbx.xfel*. Nonetheless, for the systems studied here the post-refinements converged within 3–5 cycles.

## Improvements due to post-refinement

For the myoglobin diffraction data set using all 757 images (*Table 2*, *Figure 6A,B*), the $CC_{1/2}$ value improved after post-refinement, especially for those reflections in the low-resolution shells (*Figure 5C*; *Table 2*).

Omit maps were used to compare the quality of the diffraction data processed with the different methods. Specifically, we omitted the heme group from the molecular replacement search model (PDB ID: 3U3E) and in subsequent atomic model refinement, and calculated $mF_o$-$DF_c$ difference maps (*Figure 6*). The real-space correlation coefficient of the heme group to the difference maps calculated from the post-refined diffraction data sets is higher than that calculated from the corresponding averaged merged diffraction data sets using the same set of diffraction images (*Figure 6A*).

After initial model refinement with the heme group omitted, we included the heme group and well-defined water molecules and completed the atomic model refinement. The post-refined diffraction data set produced the best $R_{free}$ and $R_{work}$ values, followed by the mean-scaled partiality corrected, with the averaged merged diffraction data sets yielding the poorest refinement statistics.

**Table 2**. Statistics of post-refinement and atomic model refinement for myoglobin

| No. images | 100 | | | 757 | | |
|---|---|---|---|---|---|---|
| **Resolution[a] (Å)** | 20.0–1.35 (1.40–1.35) | | | 20.0–1.35 (1.40–1.35) | | |
| **Completeness[a] (%)** | 80.0 (22.2) | | | 97.7 (79.8) | | |
| **Average no. observations per unique hkl[a]** | 4.0 (1.2) | | | 25.7 (2.0) | | |
| | Averaged-merged | Mean-scaled partiality corrected | Post-refined | Averaged merged | Mean-scaled partiality corrected | Post-refined |
| **Post-refinement parameters[b]** | | | | | | |
| Linear scale factor $G_0$ | 1.00 (0.00) | 2.79 (5.02) | 1.00 (1.04) | 1.00 (0.00) | 2.19 (3.83) | 0.89 (1.07) |
| $B$ | 0.0 (0.0) | 0.0 (0.0) | 3.2 (7.8) | 0.0 (0.0) | 0.0 (0.0) | 6.2 (8.3) |
| $\gamma_0$ (Å$^{-1}$) | NA | 0.00135 (0.00028) | 0.00128 (0.00022) | NA | 0.00147 (0.00042) | 0.00132 (0.00034) |
| $\gamma_y$ (Å$^{-1}$) | NA | 0.00 (0.00) | 0.00007 (0.00080) | NA | 0.00 (0.00) | 0.00007 (0.00009) |
| $\gamma_x$ (Å$^{-1}$) | NA | 0.00 (0.00) | 0.00010 (0.00011) | NA | 0.00 (0.00) | 0.00008 (0.00010) |
| $\gamma_e$ (Å$^{-1}$) | NA | 0.00200 (0.00) | 0.00344 (0.00266) | NA | 0.00200 (0.00) | 0.00423 (0.00323) |
| Unit cell | | | | | | |
| $a$ (Å): | 90.4 (0.4) | 90.4 (0.4) | 90.5 (0.4) | 90.4 (0.4) | 90.4 (0.4) | 90.5 (0.3) |
| $c$ (Å) | 45.3 (0.4) | 45.3 (0.4) | 45.3 (0.3) | 45.3 (0.3) | 45.3 (0.3) | 45.3 (0.3) |
| Average $T_{pr}$ Start/End | NA | NA | 19.39 (7.68)/7.17 (3.38) | NA | NA | 19.83 (7.54)/6.02 (2.59) |
| Average $T_{xy}$ (mm$^2$) Start/End | NA | NA | 169.74 (132.56)/132.02 (104.08) | NA | NA | 170.66 (144.52)/133.42 (109.58) |
| $CC_{1/2}$ (%) | 81.3 | 79.6 | 86.5 | 91.8 | 95.7 | 98.2 |
| **Molecular replacement scores[c]** | | | | | | |
| LLG | 2837. | 5043. | 5291. | 8264. | 8364. | 9320. |
| TFZ | 10.5 | 13.0 | 13.4 | 13.7 | 13.8 | 14.0 |
| **Structure-refinement parameters** | | | | | | |
| $R$ (%) | 39.4 | 28.0 | 23.5 | 21.1 | 20.3 | 17.8 |
| $R_{free}$ (%) | 42.1 | 29.4 | 24.8 | 23.1 | 22.5 | 19.7 |
| Bond r.m.s.d. | 0.006 | 0.006 | 0.004 | 0.006 | 0.006 | 0.006 |
| Angle r.m.s.d. | 1.14 | 0.98 | 0.79 | 1.03 | 1.35 | 0.86 |
| **Ramachandran statistics** | | | | | | |
| Favored (%) | 98.0 | 98.0 | 98.0 | 98.0 | 98.0 | 98.0 |
| Outliers (%) | 0.0 | 0.0 | 0.0 | 0.0 | 0.0 | 0.0 |

[a]Values in parentheses correspond to highest resolution shell.

[b]Post-refined parameters are shown as the mean value, with the standard deviation in parentheses.

[c]Molecular replacement scores reported by *Phaser* (**McCoy et al., 2007**): log-likelihood gain (LLG) and translation function (TFZ).

Overall, comparison of the $CC_{1/2}$ (**Figure 5**), omit map quality, and *R* values (**Figure 6B**) shows that post-refinement substantially improves scaling and correction of the diffraction data with respect to the mean-scaled partiality-corrected diffraction data set. Thus, post-refinement against the iteratively improved reference set is superior to methods that only consider each diffraction image individually, even when the reflections are scaled and corrected for partiality.

## 100 diffraction images are sufficient for myoglobin structure refinement

Given the significant improvements obtained by post-refining all available images, we tested whether accurate diffraction data and refined atomic models could be obtained using fewer diffraction images by post-refining the randomly selected subset of 100 myoglobin diffraction images. Since this subset is only 80% complete, the $CC_{1/2}$ is poorer than that of the full diffraction data set consisting of 757 images, but it is nonetheless greatly improved relative to the corresponding non-post-refined diffraction data set (**Figure 5**). Moreover, the real-space correlation coefficient of the heme group with the difference map obtained with the post-refined 100 diffraction images is better than that calculated

**Table 3**. Statistics of post-refinement and atomic model refinement for hydrogenase

| No. images | 100 | | 177 | |
|---|---|---|---|---|
| Resolution[a] (Å) | 45.0–1.60 (1.66–1.60) | | 45.0–1.60 (1.66–1.60) | |
| Completeness[a] (%) | 83.0 (47.7) | | 91.2 (63.5) | |
| Average no. observations per unique hkl[a] | 4.4 (1.7) | | 7.13 (2.3) | |
| | Averaged-merged | Post-refined | Averaged-merged | Post-refined |
| Post-refinement parameters[b] | | | | |
| Linear scale factor $G_0$ | 1.00 (0.00) | 0.56 (1.27) | 1.00 (0.00) | 0.53 (1.22) |
| $B$ | 0.0 (0.0) | 10.0 (7.0) | 0.0 (0.0) | 10.5 (6.9) |
| $\gamma_0$ (Å$^{-1}$) | NA | 0.00132 (0.00042) | NA | 0.00126 (0.00041) |
| $\gamma_y$ (Å$^{-1}$) | NA | 0.00002 (0.00004) | NA | 0.00002 (0.00004) |
| $\gamma_x$ (Å$^{-1}$) | NA | 0.00008 (0.00009) | NA | 0.00008 (0.00011) |
| $\gamma_e$ (Å$^{-1}$) | NA | 0.00269 (0.00138) | NA | 0.00288 (0.00160) |
| Unit cell | | | | |
| $a$ (Å): | 110.1 (0.4) | 110.4 (0.3) | 110.1 (0.4) | 110.3 (0.4) |
| $c$ (Å) | 103.1 (0.4) | 103.1 (0.2) | 103.0 (0.4) | 103.0 (0.2) |
| Average $T_{pr}$ Start/End | NA | 28.20 (10.86)/5.92 (2.35) | NA | 26.47 (12.70)/5.22 (2.72) |
| Average $T_{xy}$ (mm$^2$) Start/End | NA | 623.36 (314.57)/381.23 (198.44) | NA | 564.30 (267.45)/ 372.28 (202.28) |
| $CC_{1/2}$ (%) | 62.0 | 77.3 | 71.7 | 84.8 |
| Molecular replacement scores[c] | | | | |
| LLG | 53,352. | 9612. | 7229. | 11774. |
| TFZ | 69.2 | 75.9 | 75.0 | 79.0 |
| Structure-refinement parameters | | | | |
| $R$ (%) | 33.4 | 25.3 | 29.1 | 22.0 |
| $R_{free}$ (%) | 36.7 | 28.9 | 31.3 | 25.0 |
| Bond r.m.s.d. | 0.006 | 0.007 | 0.007 | 0.007 |
| Angle r.m.s.d. | 1.43 | 1.50 | 1.68 | 1.97 |
| Ramachandran statistics | | | | |
| Favored (%) | 96.3 | 97.0 | 97.0 | 96.7 |
| Outliers (%) | 0.0 | 0.0 | 0.0 | 0.0 |

[a]Values in parentheses correspond to highest resolution shell.
[b]Post-refined parameters are shown as the mean value, with the standard deviation in parentheses.
[c]Molecular replacement scores reported by Phaser (**McCoy et al., 2007**): log-likelihood gain (LLG) and translation function (TFZ).

from the averaged merged diffraction data set using all the 757 diffraction images (**Figure 6A**), despite the higher completeness and $CC_{1/2}$ value of the latter data set (**Figure 5C**). Thus, post-refinement both improves diffraction data quality for a given set of images and reduces the number of diffraction images required for structure determination and refinement from serial diffraction data.

## Comparison with a synchrotron data set

We also compared the post-refined XFEL difference map (using all 757 diffraction images) with that calculated from an isomorphous synchrotron data set and model (PDB ID: 1JW8, excluding reflections past 1.35 Å resolution to make the resolution of the diffraction data sets equivalent). The omit maps and real-space correlation coefficients for the heme group were of comparable quality (**Figure 7**).

## Hydrogenase

XFEL diffraction data for *Clostridium pasteurianum* hydrogenase were measured from eight crystals by the Peters (University of Montana) and SSRL-SMB groups using a goniometer-mounted fixed-target grid (**Cohen et al., 2014**). This experiment generated 177 diffraction images that could be merged to a completeness of 91%, with more than half of the diffraction images containing reflections to 1.6 Å

**Table 4.** Statistics of post-refinement and atomic model refinement for thermolysin

| No. images | 2000 | | 12,692 | |
|---|---|---|---|---|
| Resolution[a] (Å) | 50.0–2.10 (2.18–2.10) | | 50.0–2.10 (2.18–2.10) | |
| Completeness[a] (%) | 81.3 (24.3) | | 96.5 (74.8) | |
| Average no. observations per unique hkl[a] | 32.8 (1.2) | | 176.6 (2.4) | |
| | Averaged-merged | Post-refined | Averaged-merged | Post-refined |
| **Post-refinement parameters[b]** | | | | |
| Linear scale factor $G_0$ | 1.00 (0.00) | 1.65 (1.66) | 1.00 (0.00) | 2.26 (75.12) |
| $B$ | 0.0 (0.0) | 23.0 (33.8) | 0.0 (0.0) | 30.1 (59.8) |
| $\gamma_0$ (Å$^{-1}$) | NA | 0.00052 (0.00040) | NA | 0.00051 (0.00039) |
| $\gamma_y$ (Å$^{-1}$) | NA | 0.00001 (0.00003) | NA | 0.00001 (0.00003) |
| $\gamma_x$ (Å$^{-1}$) | NA | 0.00002 (0.00004) | NA | 0.00002 (0.00004) |
| $\gamma_e$ (Å$^{-1}$) | NA | 0.00110 (0.00129) | NA | 0.00103 (0.00128) |
| Unit cell | | | | |
| $a$ (Å): | 92.9 (0.3) | 92.9 (0.2) | 92.9 (0.3) | 92.9 (0.3) |
| $c$ (Å) | 130.5 (0.5) | 130.4 (0.4) | 130.5 (0.5) | 130.4 (0.4) |
| Average $T_{pr}$ Start/End | NA | 1.15 (0.49)/0.55 (0.23) | NA | 1.15 (0.52)/0.28 (0.13) |
| Average $T_{xy}$ (mm$^2$) Start/End | NA | 168.13 (117.29)/167.72 (106.14) | NA | 169.01 (122.20)/170.00 (122.57) |
| $CC_{1/2}$ (%) | 77.7 | 93.5 | 94.3 | 98.8 |
| **Molecular replacement scores[c]** | | | | |
| LLG | 3590. | 4491. | 5477. | 6022. |
| TFZ | 8.9 | 9.7 | 24.1 | 24.6 |
| **Structure-refinement parameters** | | | | |
| $R$ (%) | 25.2 | 19.5 | 20.7 | 18.4 |
| $R_{free}$ (%) | 29.1 | 24.0 | 23.9 | 21.1 |
| Bond r.m.s.d. | 0.004 | 0.002 | 0.002 | 0.002 |
| Angle r.m.s.d. | 0.75 | 0.58 | 0.59 | 0.62 |
| Ramachandran statistics | | | | |
| Favored (%) | 95.9 | 94.6 | 95.2 | 94.9 |
| Outliers (%) | 0.0 | 0.0 | 0.0 | 0.0 |
| Zinc peak height | | | | |
| Zn(1) (σ) | 14.0 | 16.0 | 14.3 | 20.9 |
| Zn(2) (σ) | 3.6 | 5.1 | 7.7 | 7.1 |
| Average peak height for calcium ions (σ) | 9.7 | 11.3 | 14.2 | 16.1 |

[a]Values in parentheses correspond to highest resolution shell.

[b]Post-refined parameters are shown as the mean value, with the standard deviation in parentheses.

[c]Molecular replacement scores reported by *Phaser* (**McCoy et al., 2007**): log-likelihood gain (LLG) and translation function (TFZ).

(each diffraction image typically has approximately 3000 spots). We also used a randomly selected subset of 100 diffraction images to assess the effect of post-refinement on a smaller number of images.

The $CC_{1/2}$ value improved significantly with post-refinement (**Table 3**). For quality assessment, the Fe-S cluster was omitted from both the molecular replacement search model (PDB ID 3C8Y) and subsequent atomic model refinement. The omit map densities for the post-refined diffraction data sets using the complete set of 177 diffraction images and the randomly selected subset of 100 diffraction images (83% complete) clearly show the entire Fe-S cluster whereas the densities using the averaged merged data sets are much poorer (**Figure 8A**). Upon atomic model refinement with the Fe-S clusters and water molecules included, the $R$ and $R_{free}$ values for both post-refined data sets were significantly better than the averaged merged case (**Figure 8B**).

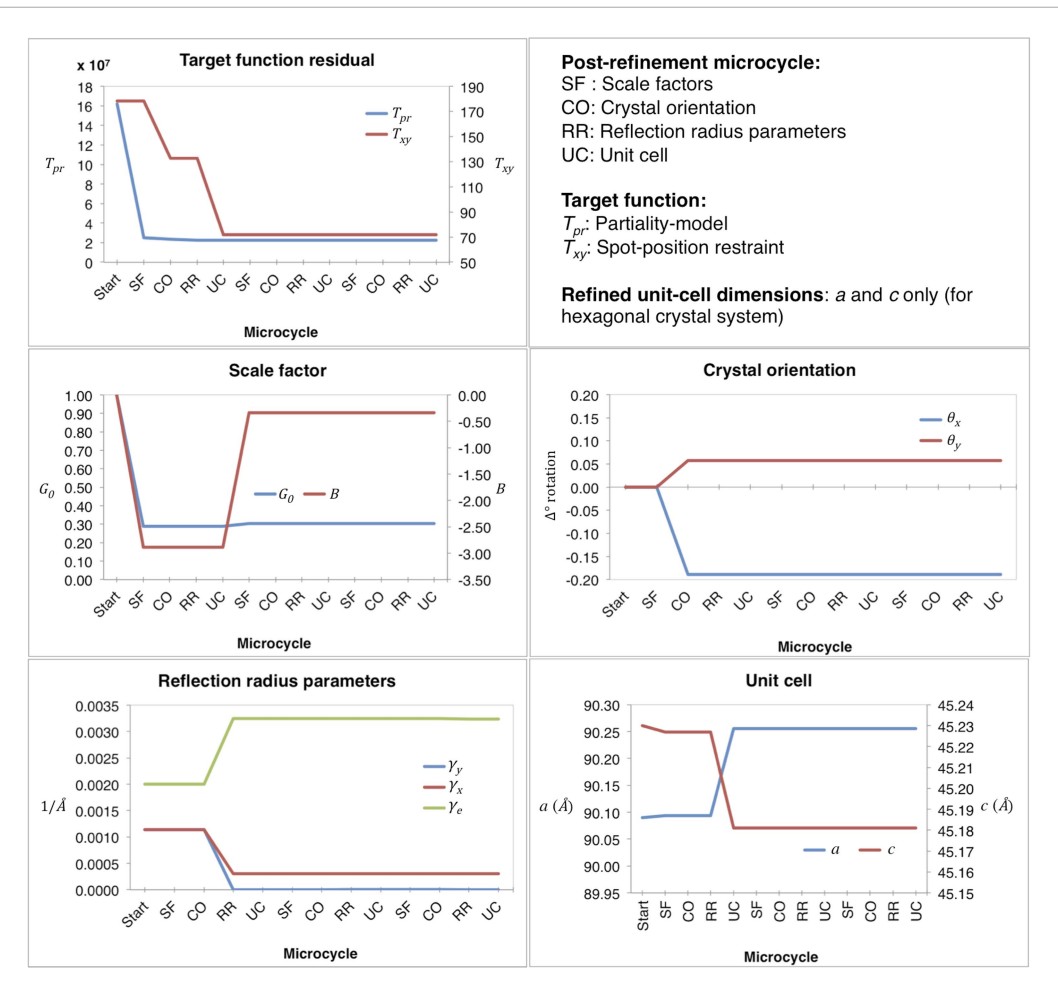

**Figure 3**. Post-refinement during the first macrocycle of post-refinement for myoglobin. Shown are the values of the refined parameters and target functions during the first macrocycle of post-refinement for a representative diffraction image of the myoglobin XFEL diffraction data set. The iterative post-refinement included SF (scale factors), CO (crystal orientation), RR (reflection radius parameters), and UC (unit-cell dimensions) for three microcycles.

## Thermolysin

For thermolysin, we tested the entire deposited XFEL diffraction data set consisting of 12,692 diffraction images (*Table 1*) (*Hattne et al., 2014*; the diffraction data are publicly archived in the Coherent X-ray Imaging Data Bank, accession ID 23, http://cxidb.org), as well as a randomly selected subset of 2000 diffraction images. In this experiment, the crystal-to-detector distance gave a maximum resolution of 2.6 Å at the edge and 2.1 Å at the corners of the detector. Thus, a large number of diffraction images were required to achieve reasonable completeness of the merged data set for reflections in the 2.1—2.6 Å resolution range.

As in the other two cases, post-refinement significantly improved the $CC_{1/2}$ value (*Table 4*). For quality assessment, zinc and calcium ions were omitted from the thermolysin molecular replacement search model (PDB ID: 2TLI) and subsequent atomic model refinement. Post-refinement improved the peak heights of both the zinc and calcium ions (*Table 4*).

### Anomalous difference Fourier peak heights

The thermolysin diffraction data were collected at a photon energy just above the absorption edge of zinc, so we compared the anomalous signals with and without post-refinement. We used the same

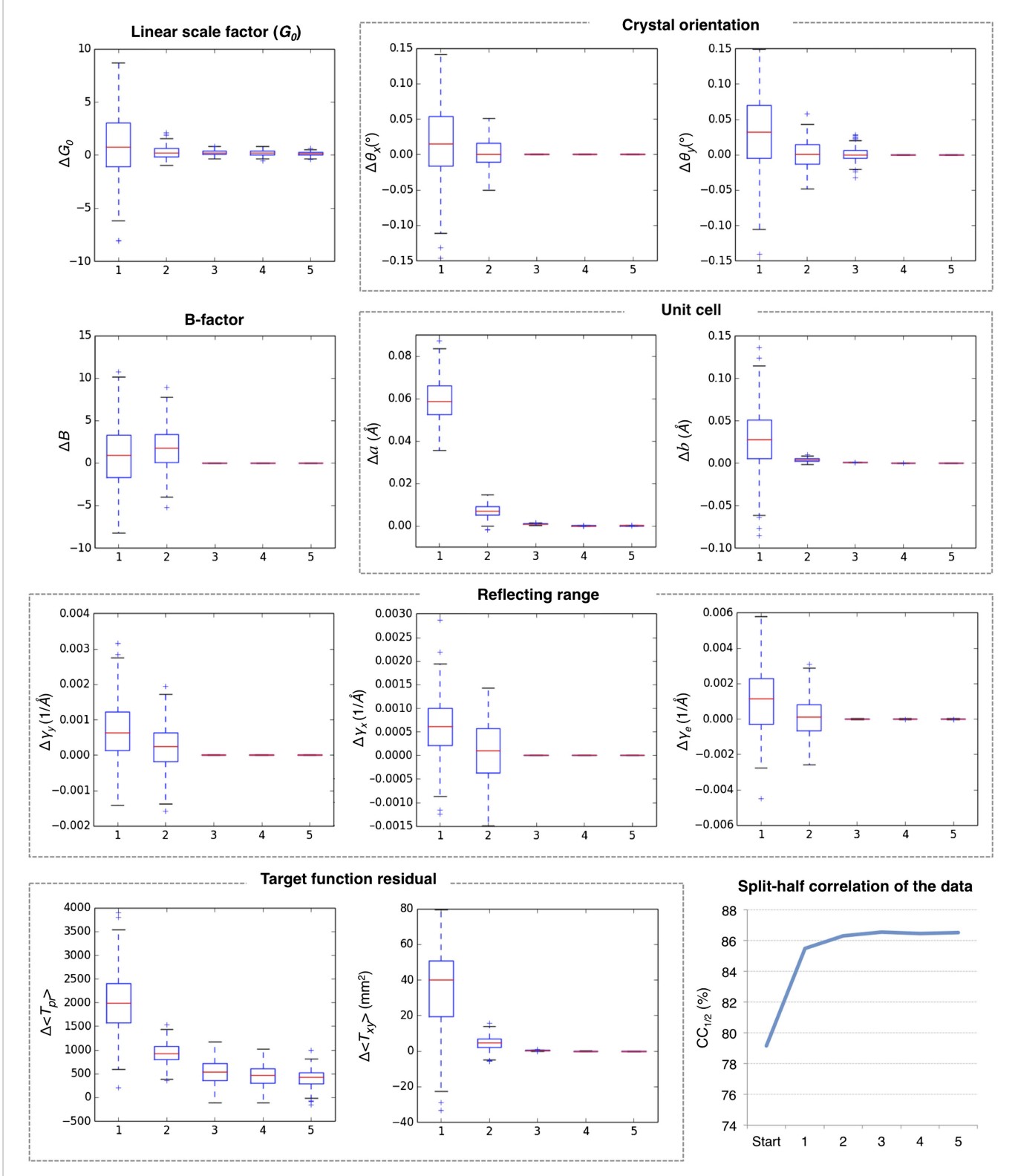

**Figure 4**. Convergence of post-refinement after five macrocycles for myoglobin. The plots illustrate the convergence of post-refined parameters, target functions, and quality indicators during post-refinement over five macrocycles. A subset of 100 randomly selected diffraction images from the myoglobin XFEL diffraction data was used. For each specified target function and refined parameter, changes are plotted relative to the previous macrocycle, whereas the quality metric $CC_{1/2}$ is shown as absolute numbers. The changes in post-refined parameters and target functions are shown as 'box plots'.
*Figure 4. continued on next page*

Figure 4. Continued

The bottom and top of the blue box are the first (Q1) and third (Q3) quartiles. The red line inside the box is the second quartile (Q2; median). The black horizontal lines extending vertically from the box indicate the range of the particular quantity at a 1.5 interquartile range (Q3–Q1). The plus signs indicate any items beyond this range.

four diffraction data sets (i.e., averaged-merged, post-refined, with 2000 and 12,692 diffraction images, respectively), but processed them keeping Friedel mates separate. We refined the atomic model of thermolysin lacking zinc and calcium ions, and calculated anomalous difference Fourier maps (*Figure 9*). We observed two anomalous difference peaks near the active site above 3 σ using the post-refined data sets. In contrast, the second, smaller peak is not visible in the anomalous difference map using the 'averaged-merged' data set with 2000 images, and it had not been clearly visible in the previous data analysis of the thermolysin XFEL data set (PDB ID: 4OW3; *Hattne et al., 2014*). A previous thermolysin structure (PDB ID: 1LND; *Holland et al., 1995*) reported two zinc sites in the active site that correspond to the two anomalous-difference peaks observed with our post-refined data set. Although the crystallization condition used in our case did not have the high concentration (10 mM) of zinc used in the Holland et al. study, the second anomalous difference peak suggests the presence of this second zinc site.

### Difference map reveals a bound dipeptide

When the molecular replacement model of thermolysin was refined against the post-refined data, we observed a well-connected electron density feature in the $mF_o$-$DF_c$ map near the active site. In contrast, in the deposited model refined against the original XFEL data (*Hattne et al., 2014*; PDB ID: 4OW3), weak density features in this region were interpreted as water molecules. We found several examples of deposited thermolysin structures that have a dipeptide in this region (e.g., PDB entry 2WHZ with Tyr–Ile, PDB entry 2WI0 with Leu–Trp, and PDB entry 8TLN with Val–Lys). We interpreted the shape of the difference density as a Leu–Lys dipeptide, superimposed its structure and calculated real-space correlation coefficients. The dipeptide had a higher real-space correlation coefficient (CC) with the maps calculated from the post-refined diffraction data than those calculated from the averaged merged diffraction data. The electron density for both post-refined diffraction data sets is also better connected than that of the averaged merged diffraction data set (*Figure 10A*). The $R_{work}$ and $R_{free}$ values of the refined complete model using the post-refined diffraction data are lower than those using the averaged merged data throughout the entire resolution range (*Figure 10B*).

## Effect of completeness

The completeness of the merged data sets has a direct impact on the overall quality of the diffraction data set ($CC_{1/2}$), quality of the electron density maps and the refined structures (*Tables 2–4*, and *Figure 6*). When completeness is high, adding more images to increase the multiplicity of observations has only a modest impact on the quality of the final refined structures using the post-refined diffraction data. For example, when subsets ranging from 2000 to 12,000 thermolysin diffraction images (all subsets 100% complete at 2.6 Å) were post-refined the peak height in the omit map for the larger of the two anomalous sites (*Figure 11C*), the $CC_{1/2}$ values, and the $R$ values of the refined structures did not improve significantly when more than 8000 images were used.

## Discussion

Diffraction data collection using conventional x-ray sources typically employs the rotation method, in which a single crystal is rotated through a contiguous set of angles, and the diffraction patterns are recorded on a 2-D detector. If a full data set can be collected from a single crystal without a prohibitive level of radiation damage, diffraction data processing is a well-established and reliable process. In contrast, processing of XFEL diffraction data, which are collected from crystals in random orientations as 'still' diffraction images, requires new methods and implementations such as those described here. Improved data collection and processing methods, particularly those that can significantly reduce the amount of sample needed to assemble a complete and accurate diffraction data set, are important for making XFELs useful for certain challenging investigations in structural biology.

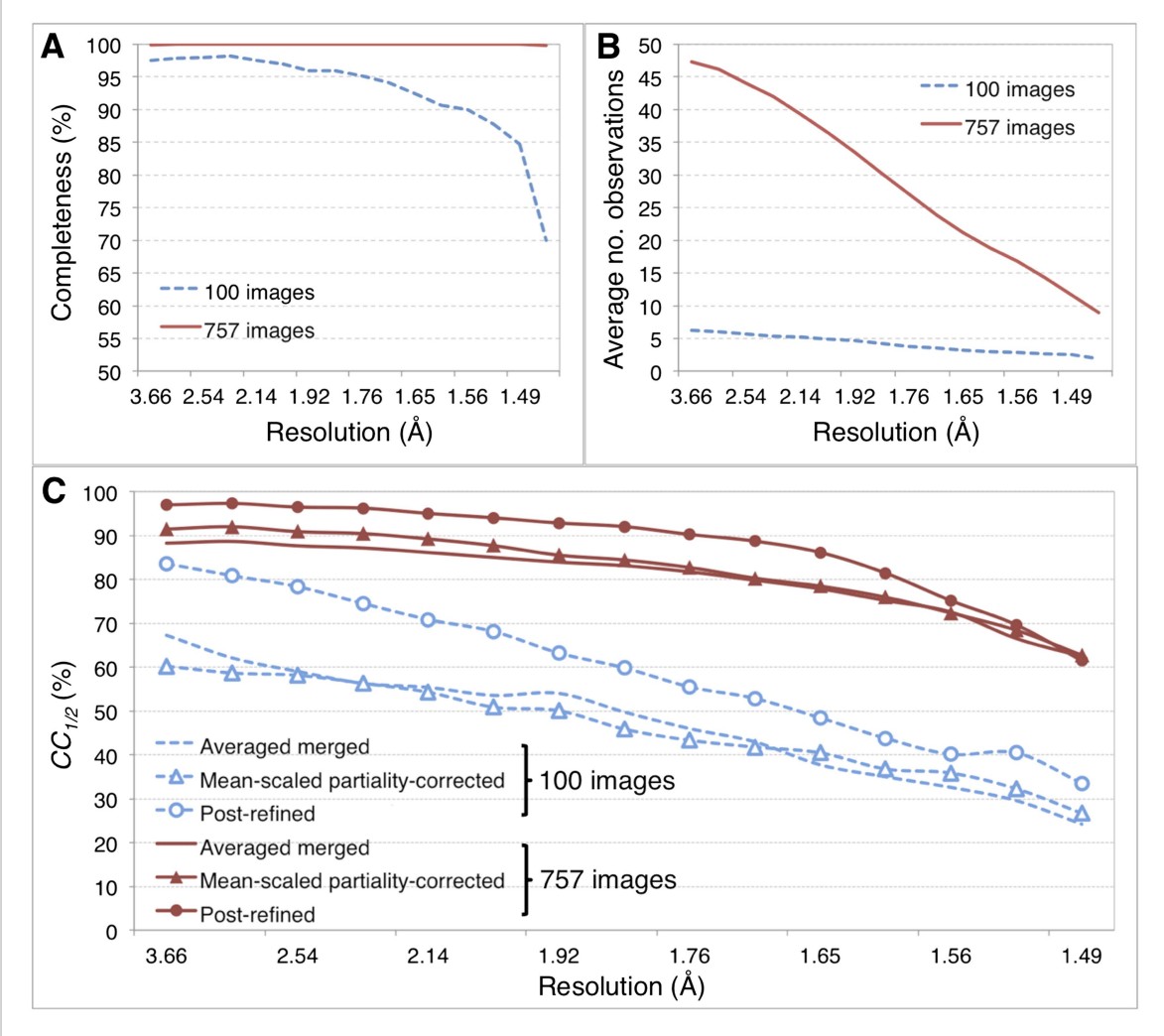

**Figure 5**. Merging statistics for myoglobin. (**A**) Percent completeness and (**B**) average number of observations plotted as a function of resolution for the myoblogin XFEL diffraction data set consisting of all 757 diffraction images (*Table 1*) and a randomly selected subset of 100 diffraction images. (**C**) $CC_{1/2}$ for the averaged merged, mean-intensity scaled with partiality correction, and post-refined myoglobin diffraction data sets consisting of 100 and 757 diffraction images.

We developed a post-refinement method for still diffraction images, such as those obtained at XFELs, and implemented it in new computer program, *prime*, that applies a least-squares minimization method to refine parameters as defined in our partiality model. Other post-refinement methods for XFEL diffraction data have been described recently (*Kabsch, 2014*; *White, 2014*), but our implementation differs from these reports. Kabsch uses a partiality model in which an Ewald offset correction is defined as a Gaussian function of angular distance from the Ewald sphere. White used the intersecting volume between the reflection and the limiting-energy Ewald spheres defined by the energy spectrum for the partiality calculation, and calculates the initial reference data set by averaging all observations without scaling. Neither report describes an application to experimental XFEL diffraction data, so we cannot compare these methods to the results presented here.

We have demonstrated here that our implementation of post-refinement substantially improves the quality of the diffraction data from three different XFEL experiments. Moreover, the resulting structures can be refined to significantly lower $R_{free}$ and $R$ values, with electron density maps that reveal novel features more clearly, than those using non-post-refined XFEL data sets. A key feature of our method is that the parameters that define the diffracted spot are iteratively refined against the

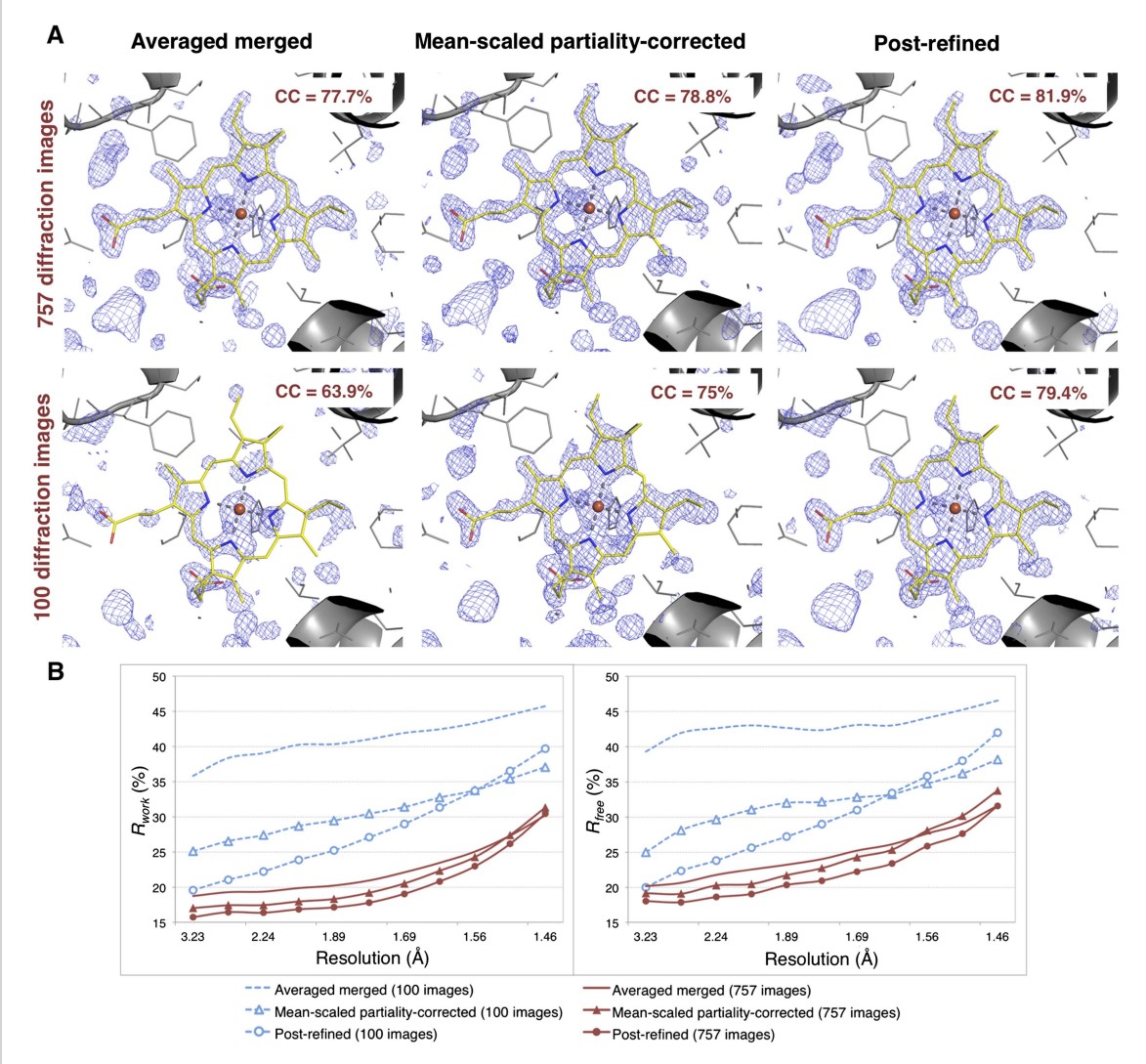

**Figure 6**. Impact of post-refinement and number of images on electron density and model quality for myoglobin. (**A**) Difference Fourier ($mF_o$-$DF_c$) omit maps around the heme group (which was omitted from molecular replacement and atomic model refinement) for the averaged merged, the mean-scaled partiality-corrected merged, and the post-refined myoglobin XFEL diffraction data sets consisting of all 757 diffraction images (*Table 1*) and a randomly selected subset of 100 diffraction images. The maps are contoured are at 2.5 σ. (**B**) A plot of crystallographic $R_{work}$ and $R_{free}$ values vs resolution after atomic model refinement using the specified myoglobin diffraction data sets with inclusion of the heme group, $SO_4$, and water molecules.

reference set. This approach is superior to methods that only consider each diffraction image individually. Moreover, our post-refinement procedure allows accurate diffraction data sets to be extracted from a much smaller number of images (average number of observations) than that necessary without post-refinement. Thus, this development will make XFEL crystallography accessible to many challenging problems in biology for which sample quantity is a major limiting factor.

At present, it is difficult to assess the relative quality of post-refined XFEL data studied here with conventional rotation data measured at a synchrotron. The comparison of myoglobin omit maps (*Figure 7*) suggests that the SR data are perhaps somewhat better, but more systematic studies will be needed to understand the relative merits of the different data sets. We suspect that rotation data would be better due to the ability to directly measure full reflections (at least by summation of partials) without modeling partiality, which is still a relatively crude process (see below). However, a comparison between still data sets measured at a synchrotron and an XFEL is needed to deconvolute the effect of rotation vs other differences between these sources.

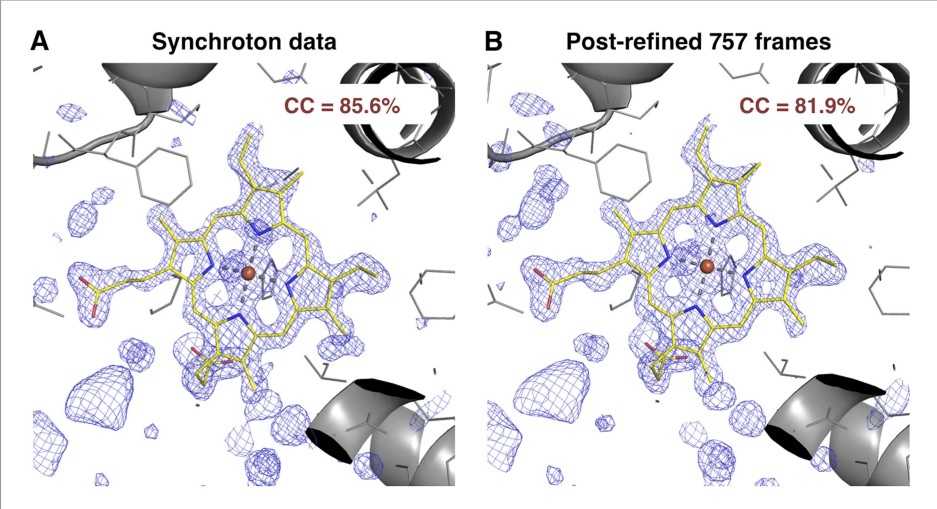

**Figure 7**. Quality of synchrotron vs. post-refined XFEL diffraction data sets for myoglobin. Difference Fourier ($mF_o$-$DF_o$) omit maps at 1.35 Å around the heme group (which was omitted from molecular replacement and model refinement), generated from (**A**) the synchrotron diffraction data and corresponding model with PDB ID 1JW8 (for comparison, all reflections past 1.35 Å resolution were excluded) and (**B**) the post-refined myoglobin XFEL diffraction data set using all 757 diffraction images (*Table 1*). The maps are contoured at 2.5 σ.

Our formulation of post-refinement employs the simplifying assumption that reflections are spherical volumes. More sophisticated models consider crystal mosaicity to have three components, each with a distinct effect on the reciprocal lattice point (*Juers et al., 2007*; *Nave, 1998*, *2014*). First, the domain size (the average size of the coherently scattering mosaic blocks) produces reciprocal lattice points of constant, finite size: small domains produce large-sized spots, while large domains produce small spots, as there is an inverse (Fourier) relation between spot size and domain size. Second, unit-cell variation among domains produces reflections that are spheres whose radii increase with distance from the origin. *In cctbx.xfel*, mosaicity (modeled as isotropic parameter) and effective domain size are taken into account when predicting which reflections are in diffracting position prior to integration (*Sauter et al., 2014*; *Sauter, 2015*). Third, orientational spread among mosaic domains produces spots shaped like spherical caps. Each cap subtends a solid angle that depends on the magnitude of the spread. In addition, anisotropy in crystal mosaicity is not considered; this would require refining separate parameters along each lattice direction. Finally, the rugged energy spectrum that results from the SASE process of the XFEL is not yet considered in our current model. These issues will require future investigation.

## Materials and methods

### Partiality model

The observed intensity $I_h(i)$ for observation $i$ of Miller index **h** is a thin slice through a three-dimensional reflection. To calculate partiality, we assume that the measurement is an infinitely thin, circular sample of a spherical volume (*Figure 1B*). We assume a monochromatic beam as the starting point to define the Ewald offset correction $Eoc_{area}$. The $Eoc_{area}$ of any reflection centered on the Ewald sphere is defined as 1; this position corresponds to the maximum partial intensity that could be measured for the reflection. The $Eoc_{area}$ for any other position is defined as a function of the normal distance from the Ewald sphere to the center of the reciprocal lattice point (the offset distance, $r_h$), and of the reciprocal-lattice radius of the spot $r_s$, which is a function of the crystal mosaicity and spectral dispersion (*Figure 1B*). The $Eoc_{area}$ can be described by the ratio of the observed area ($A_p$) with a radius $r_p$ to the Ewald-offset corrected area ($A_s$) with a radius $r_s$ (*Figure 1B*).

The SASE spectrum emitted by the XFEL is broad and varies from shot-to-shot (*Zhu et al., 2012*). To calculate the Ewald sphere, we set the wavelength to be the centroid of the SASE spectrum

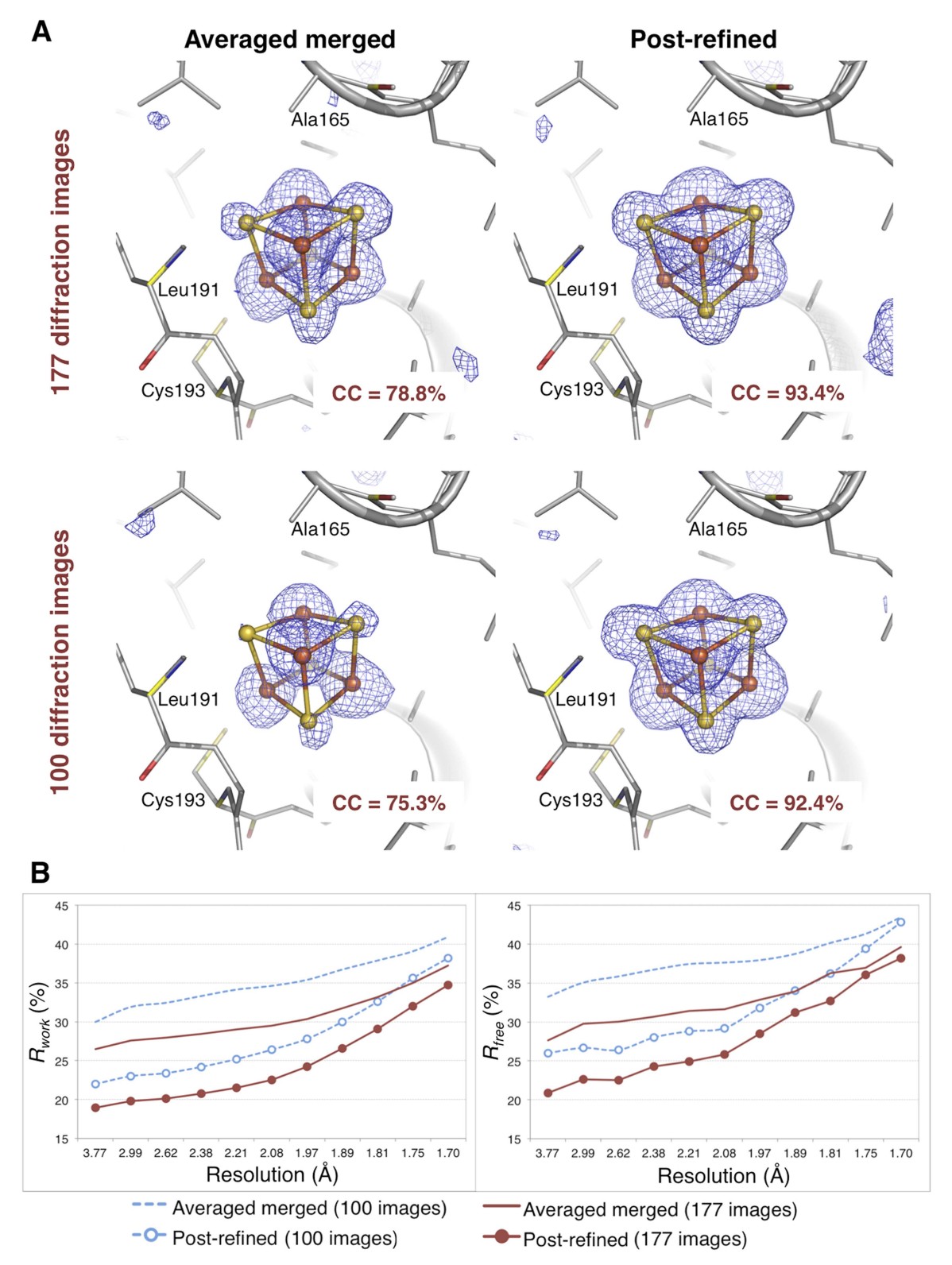

**Figure 8**. Impact of post-refinement on the hydrogenase diffraction data set. (**A**) Difference Fourier ($mF_o$-$DF_c$) omit maps of one of the four Fe-S clusters (which were omitted in molecular replacement and atomic model refinement) for the averaged merged and the post-refined hydrogenase XFEL diffraction data sets consisting of all 177 diffraction images (**Table 1**) and a randomly selected subset of 100 diffraction images. The maps are contoured at 3 $\sigma$. (**B**) Crystallographic $R$ and $R_{free}$ values vs resolution after atomic model refinement using the specified diffraction data sets with inclusion of the three Fe-S clusters and water molecules.

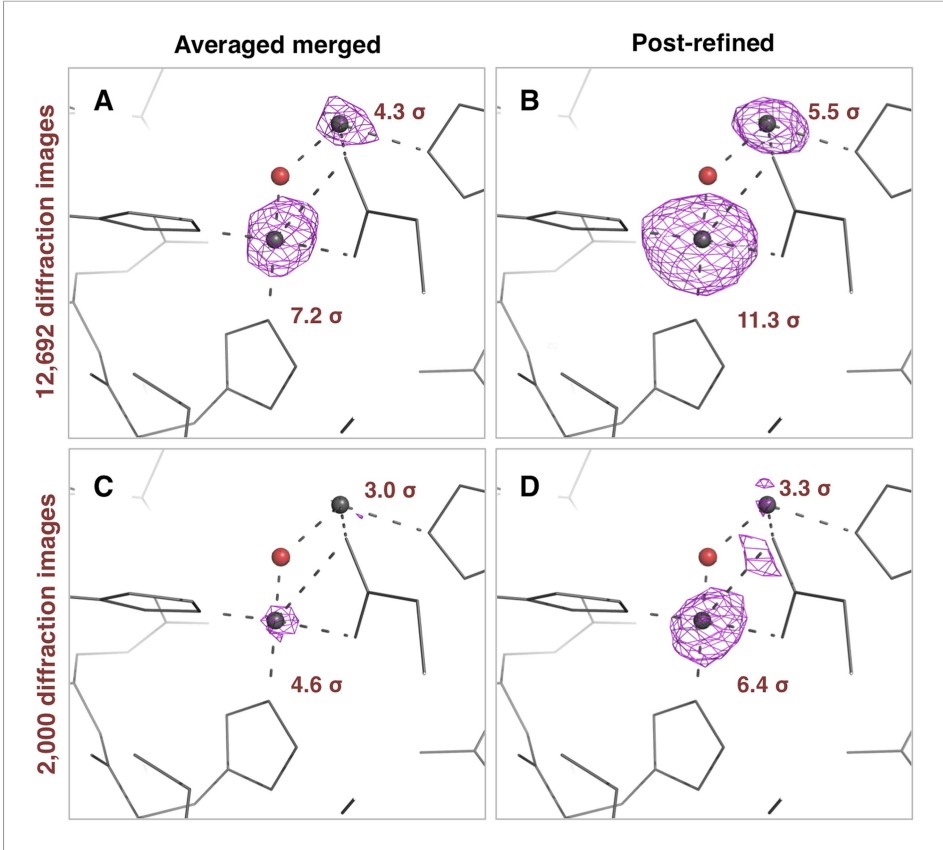

**Figure 9**. Impact of post-refinement on the anomalous signal in the thermolysin diffraction dataset. Anomalous difference Fourier maps for the averaged merged (**A**, **C**) or the post-refined (**B**, **D**) thermolysin XFEL diffraction data sets consisting of all 12,692 diffraction images (**A**, **B**—*Table 1*) and a randomly selected subset of 2000 diffraction images (**C**, **D**). The anomalous difference Fourier maps were computed using phases from the thermolysin atomic model (but excluding zinc and calcium ions), refined separately against each diffraction data set. All maps are contoured at 3 $\sigma$; the peak heights for the two zinc ions are indicated.

recorded with each shot. For XFEL data measured with a seeded beam (*Amann et al., 2012*), the spectrum is narrow and constant from shot-to-shot, and this single value can be used in this case.

In order to model spectral dispersion and the possible effects of asymmetric beam divergence, we adapt the rocking curve model described in *Winkler et al. (1979)*. The four-parameter function used for the rocking curve is $r_s(\gamma_0, \gamma_e, \gamma_x, \gamma_y) = r_s(\theta) + r_s(\alpha)$, where the first term includes the contribution by spectral dispersion and the second term models beam anisotropy. Specifically,

$$r_s(\theta) = \gamma_0 + \gamma_e \tan \theta, \tag{3}$$

where $\gamma_0$ is a parameter that is initially set to the r.m.s.d. of the Ewald offset calculated for all the reflections on a given image, $\gamma_e$ represents the width of the energy spread and the unit-cell variation (the initial value of $\gamma_e$ is calculated from the average energy spread), and $\theta$ is the Bragg angle. The second term is provided by:

$$r_s(\alpha) = \left[ \left( \gamma_y \cos \alpha \right)^2 + \left( \gamma_x \sin \alpha \right)^2 \right]^{1/2}, \tag{4}$$

where $\alpha$ is the azimuthal angle going from meridional ($\alpha = 0$) to equatorial ($\alpha = \pi/2$). The values of $\gamma_y$ and $\gamma_x$ are initially set to 0.

The distribution of $r_h$ values for the myoglobin case with 757 images after post-refinement is shown in *Figure 12*. The parameters $\gamma_e$, $\gamma_y$, $\gamma_x$, $\gamma_0$ are refined within a microcycle (*Figure 2*).

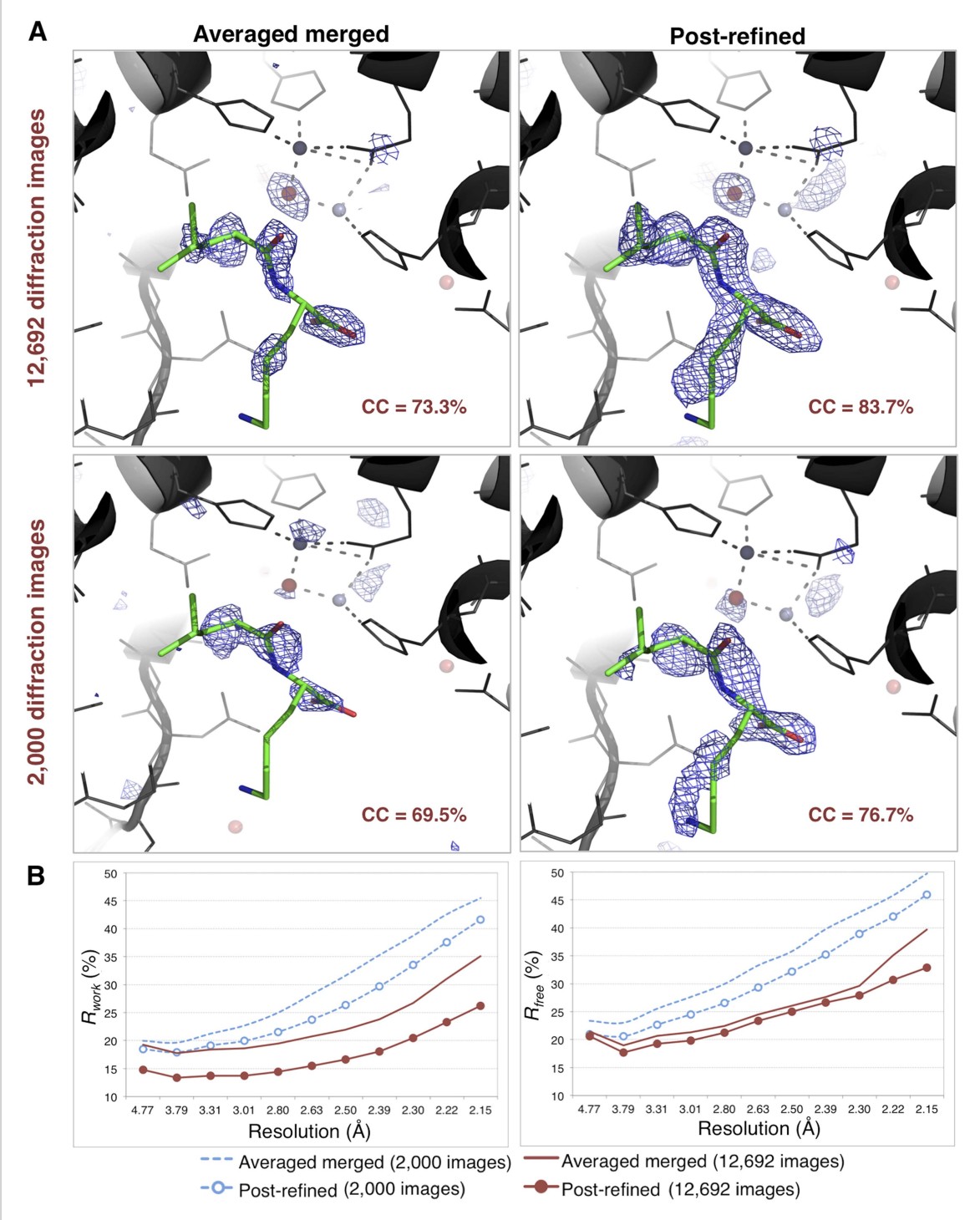

**Figure 10**. Impact of post-refinement on the quality of electron density maps and models of thermolysin. (**A**) Difference Fourier ($mF_o$-$DF_c$) maps revealing a Leu–Lys dipeptide near the zinc site for the averaged merged and the post-refined thermolysin XFEL diffraction data sets consisting of all 12,692 diffraction images (*Table 1*) and a randomly selected subset of 2000 diffraction images, respectively. The maps are contoured at 3 $\sigma$. (**B**) Crystallographic $R$ and $R_{free}$ values vs resolution for the refinements after atomic model refinement using the specified diffraction data sets and with inclusion of two zincs, calcium ions, and the Leu–Lys dipeptide.

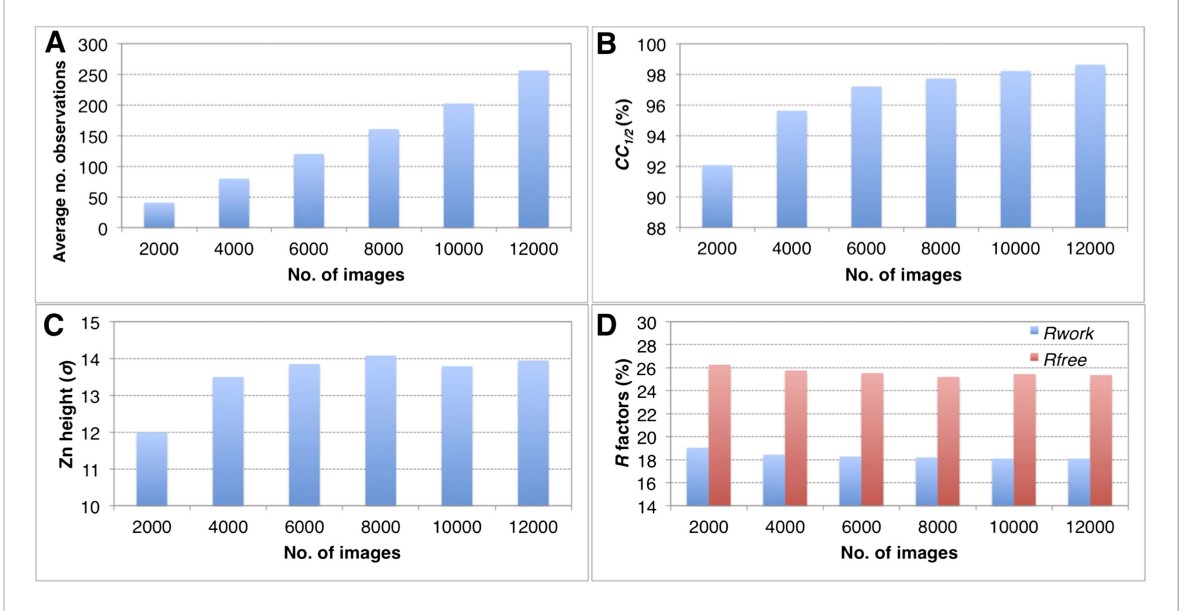

**Figure 11**. Convergence of structure refinements for the post-refined thermolysin XFEL data set at 2.6 Å resolution, using increasing numbers of diffraction images. (**A**) Average number of observations per unique hkl. (**B**) $CC_{1/2}$ for merged subsets using 2000–12,000 images (100% completeness for all subsets). (**C**) Peak height (σ) in the omit map for the largest peak. (**D**) $R_{work}$ and $R_{free}$ after refining the thermolysin model without zinc and calcium ions against the corresponding post-refined diffraction data sets.

## Calculating the reciprocal lattice point offset

The crystal orientation is described in a right-handed coordinate system with the *z*-axis pointing to the source of the incident beam and the y-axis vertical (***Figure 1A***). We define the crystal orientation by rotations in the order $\theta_z$, $\theta_y$, $\theta_x$ about these axes. For each Miller index $h(i)$, the reciprocal lattice point vector $x(i)$ is obtained by applying orthogonalization and rotation matrixes $O$ and $R$:

$$x(i) = ROh(i), \tag{5}$$

where

$$x(i) = (x(i), y(i), z(i)),$$

$$h(i) = (h(i), k(i), l(i)),$$

$$O = \begin{pmatrix} a^* & b^* \cos \gamma^* & c^* \cos \beta^* \\ 0 & b^* \sin \gamma^* & c^*(\cos \alpha^* - \cos \gamma^*)/\sin \gamma^* \\ 0 & 0 & c^* \cos(c^*, c) \end{pmatrix},$$

$$R = R_{\theta_x} R_{\theta_y} R_{\theta_z},$$

where $R_{\theta_i}$ is the rotation matrix for a rotation around the i-th axis, $a^*$, $b^*$, $c^*$, $\alpha^*$, $\beta^*$, $\gamma^*$ are the reciprocal unit-cell parameters, and $\cos(c^*, c) = (1 + 2 \cos \alpha^* \cos \beta^* \cos \gamma^* - \cos^2 \alpha^* - \cos^2 \beta^* - \cos^2 \gamma^*)^{1/2}/\sin \gamma^*$.

As shown in ***Figure 1A***, the displacement to $x(i)$ from the center of the Ewald sphere is given by:

$$S(i) = x(i) + S_0, \tag{6}$$

where $S_0 = (0, 0, -1/\lambda)$. The offset distance is thus the difference between the length of $S(i)$ and the Ewald-sphere radius,

$$r_h = |S(i)| - 1/\lambda. \tag{7}$$

## The Ewald-offset correction function Eoc

We introduce a smooth approximation of the area ratio $Eoc_{area}$ (see 'Results') in order to circumvent the undefined first derivative when the ratio is zero. We use a Lorentzian function ($f_L$) to model the radius as function of distance from the Ewald sphere:

$$f_L = \frac{1}{\pi} \frac{\frac{1}{2}\Gamma}{(r_h)^2 + \left(\frac{1}{2}\Gamma\right)^2}. \tag{8}$$

The function is normalized so that $f_L(r_h = 0) = 1.0$ when the reciprocal-lattice point is centered on the Ewald sphere, so that

$$f_{Ln} = \frac{\pi\Gamma}{2} f_L. \tag{9}$$

We then use the ratio of the observed area ($A_p$) with a radius $r_p$ to the Ewald-offset corrected area ($A_s$) with a radius $r_s$ (**Figure 1B**) that corresponds to the full width at half maximum (FWHM), $\Gamma$, in the Lorentzian function. Using the Lorentzian function to describe the falloff in radius as we move away from the Ewald sphere makes the $Eoc$ function differentiable at $r_h = r_s$. For the reciprocal lattice volume being bound by a sphere of radius $r_s$ centered on the reciprocal lattice point, the intersecting area of the volume is given by:

$$A_p = \pi r_p^2, \tag{10}$$

where

$$r_p = \left(r_s^2 - r_h^2\right)^{1/2}.$$

The $Eoc$ is then given by the ratio of this intersecting area to the area when this reflection is centered on the Ewald sphere ($A_s$),

$$\boldsymbol{Eoc}_{area} = \frac{A_p}{A_s} = \frac{\pi r_p^2}{\pi r_s^2} = 1 - \frac{r_h^2}{r_s^2}. \tag{11}$$

By setting the FWHM of $\Gamma$ proportional to the radius, $r_s$, at half $Eoc_{area}$,

$$\boldsymbol{Eoc}_{area} = 1 - \frac{r_h^2}{r_s^2} = 0.5, \tag{12}$$

$$\Gamma = r_s \text{ at } 0.5 \ \boldsymbol{Eoc}_{area} = \sqrt{2} r_h, \tag{13}$$

we arrive at the Ewald-offset correction function (**Figure 13A**)

$$\boldsymbol{Eoc} = \frac{r_s^2}{2r_h^2 + r_s^2}. \tag{14}$$

The use of this Lorentzian approximation to derive the $Eoc$ function vs an actual sphere function, $Eoc_{area}$, is illustrated in **Figure 13B**.

## Correction to full intensity

To adjust the observed still intensity to its equivalent at zero offset, we apply the Ewald-offset correction to the observed intensity,

$$I_{\boldsymbol{Eoc},h}(i) = \frac{I_h(i)}{\boldsymbol{Eoc}_h(i) G_m}, \tag{15}$$

where $I_h(i)$ is the observed partial intensity $i$ of Miller index $\boldsymbol{h}$ on image $m$, $\boldsymbol{Eoc}_h(i)$ is the Ewald-offset correction, and $G_m$ is a scale function for image $m$. We then convert this maximum partial intensity to a full intensity estimate by correcting for the volume of the spot, a factor of $\frac{\frac{4}{3}\pi r_s^3}{\pi r_s^2} = \frac{4}{3} r_s$:

$$I_{full,h}(i) = V_{c,h}(i) I_{Eoc,h}(i), \tag{16}$$

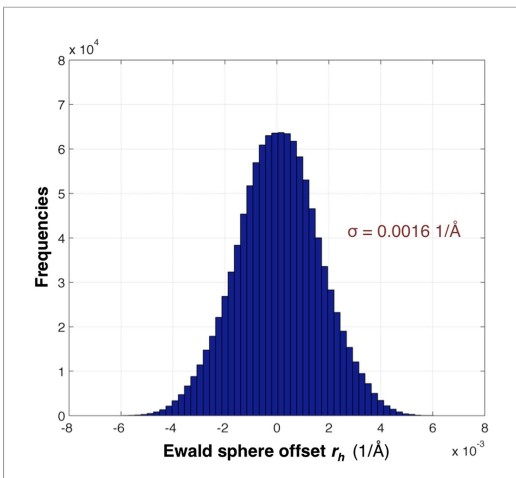

**Figure 12**. Distribution of the Ewald sphere offset $r_h$. The histogram shows the distribution of $r_h$ calculated after post-refinement for myoglobin using 757 diffraction images. The number of observations after applying the reflection selection criteria for merging and outlier rejections for this 1.35 Å data set is 1,136,447 (~96% of the total observed reflections). The standard deviation is 0.0016 1/Å or approximately 0.12° (when calculated with the mean of the energy distribution).

where

$$V_{c,h}(i) = \frac{4}{3} r_{s,h}(i).$$

Note that $I_{full,h}(i)$ will be on an arbitrary scale, and appropriate scaling methods may be applied to place the data on a quasi-absolute scale prior to structure determination and refinement, as is done for conventional rotation data.

## Refinement of crystal orientation, reflection width, and unit-cell parameters

We refine image $m$ by first minimizing the target function:

$$T_{pr} = \sum_h \sum_i W_h(i) \left( I_h(i) - G_m \boldsymbol{Eoc}_h(i) V_{c,h}^{-1}(i) I_h \right)^2,$$

(17)

where

$$1/W_h(i) = \sigma_h^2(i),$$

and the scale function $G_m$ comprises a linear scale factor $G_0$ and a $B$-factor:

$$G_m = G_{0,m} e^{-2B_m(\sin \theta_h(i)/\lambda_m)^2}.$$

(18)

We apply a spot position restraint as a second target function in subsequent steps during a microcycle using the $x$, $y$ positions determined by the spot-finding step of data processing (*Hattne et al., 2014*; *Kabsch, 2014*).

$$T_{xy} = \sum_h \sum_i \left( x_h^{obs}(i) - x_h^{calc}(i) \right)^2,$$

(19)

where $x_h^{obs}(i)$ and $x_h^{calc}(i)$ are the observed and calculated spot centroids, respectively.

The Levenberg–Marquardt (LM) algorithm from the *scipy* python library (*Oliphant, 2007*), which is a combination of the gradient descent and the Gauss–Newton iteration, is used to minimize the target function residuals. The refinement of the unit-cell parameters ($a$, $b$, $c$, $\alpha$, $\beta$, $\gamma$) takes crystal symmetry constraints into account to make the procedure more robust.

After these iterative refinement cycles are complete, we apply the refined parameters to the reflection intensities of each still, and then merge the same reduced Miller indices (from all stills) to obtain the zero-offset still intensities, which are used for the new reference intensity set (see next section).

### Reflection selection criteria

At each step in a microcycle, the user can select reflections that are used for post-refinement of a parameter group using the following criteria: resolution range, signal strength ($I/\sigma(I)$), and the Ewald offset correction value. In addition to these selection criteria, deviations from the target unit-cell dimensions (specified as a fraction of each dimension) can also be used in the merging step so that only diffraction patterns with acceptable unit-cell dimension values are included in the merged reflection set. Each post-refinement parameter group can have its own separate set of reflection selection criteria.

### Merging procedure

Starting from the observed intensities, we obtain the full-volume intensity, $I_{full,h}(i)$, from $I_{Eoc,h}(i)$ by first applying the Ewald offset correction (*Equation 15*) and then the full-intensity correction (*Equation 16*). Prior to merging equivalent observations, we detect outliers using an iterative rejection scheme, discarding reflections with intensity more or less than a user-specified cutoff (3 σ default, where σ is defined as the standard deviation of the distribution of the full reflections $I_{full,h}$). Finally, in order to

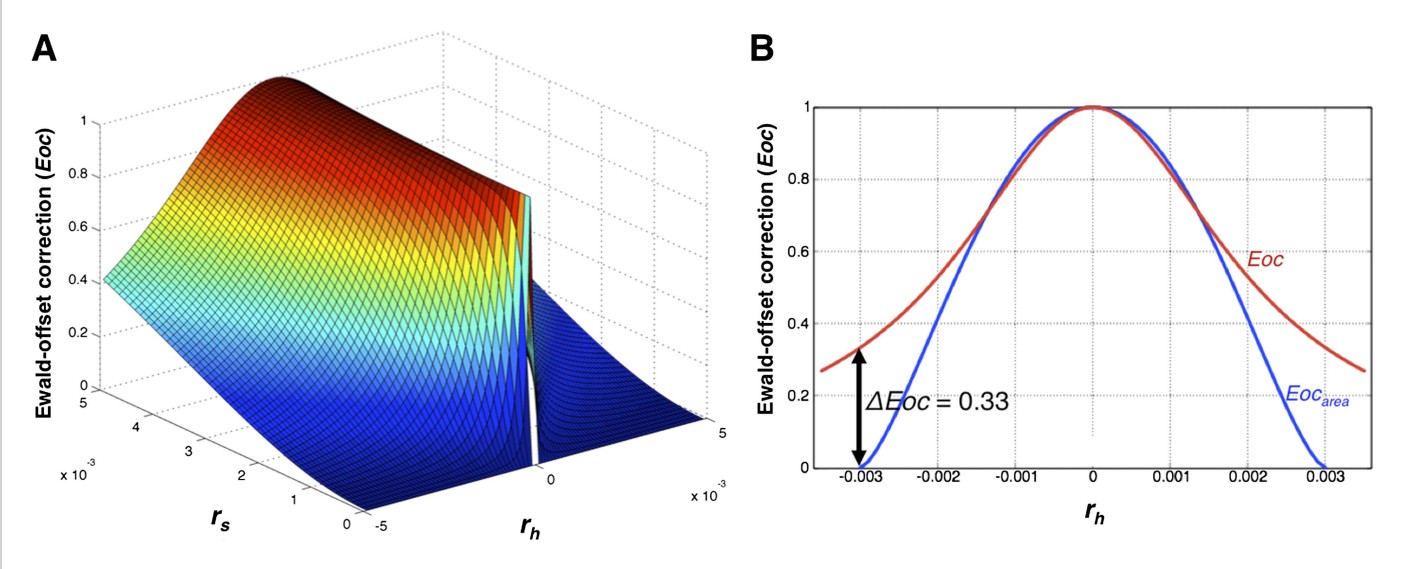

**Figure 13**. The Ewald-offset correction function. (**A**) Ewald-offset correction $Eoc$ (**Equation 14**) viewed as a function of the reciprocal-lattice radius ($r_s$) and the offset distance ($r_h$). (**B**) A slice through $Eoc$ at $r_s = 0.003$, comparing $Eoc$ (**Equation 14**) and $Eoc_{area}$ (**Equation 11**).

obtain the merged reflection set, we calculate $\langle I_h \rangle$ from the intensity of reflections with the same reduced Miller indices using the sigma-weighted average:

$$\langle I_h \rangle = \frac{\sum_i W_h(i) I_{full,h}(i)}{\sum_i W_h(i)}, \tag{20}$$

where

$$W_h(i) = \frac{1}{\sigma^2(i)\left[ I_{full,h}(i) \right]}, $$

and $\sigma(i)[I_{full,h}(i)]$ is derived from the calculation of error:

$$\left( \frac{\Delta I_{full,h}(i)}{I_{full,h}(i)} \right)^2 = \left( \frac{\Delta I_h(i)}{I_h(i)} \right)^2 + \left( \frac{\Delta G}{G} \right)^2 + \left( \frac{\Delta Eoc}{Eoc} \right)^2. \tag{21}$$

Since $G$ is a function of $G_0$ and $B$, and $Eoc$ is a function of crystal orientation, mosaicity, and unit-cell parameters, the error estimates for $G$ can be further calculated as:

$$\Delta G^2 = \left( \frac{\partial G}{\partial G_0} \right)^2 \Delta G_0^2 + \left( \frac{\partial G}{\partial B} \right)^2 \Delta B^2, \tag{22}$$

and $\Delta Eoc^2$ can be calculated similarly by summing all over products of partial derivatives and errors estimated for each parameter in the $Eoc$ function (square root of the diagonal elements of the covariance matrix).

We use $CC_{1/2}$ as a quality indicator for the diffraction data sets (**Diederichs and Karplus, 2013**). We calculate $CC_{1/2}$ by randomly partitioning all (partial) intensity observations of a given reflection into two groups. We reject any reflections with fewer than four observations; for all other reflections, we merge the observations in each group using **Equation 20**. $CC_{1/2}$ is then calculated as the correlation between these two independently merged diffraction data sets.

## Partial derivatives of the diffraction parameters

Let

$$g = \frac{1}{\sigma} \left( I - G_0 e^{-2B(\sin\theta/\lambda)^2} Eoc V_c^{-1} I \right), \tag{23}$$

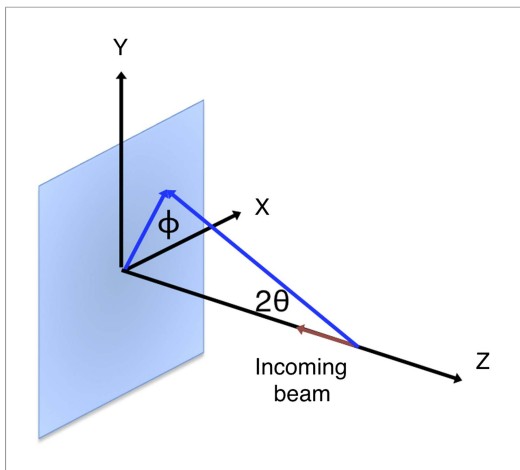

**Figure 14**. Geometry of the incident and diffracted beam for polarization correction. The diagram shows a reflection on a plane formed by its reciprocal-space vector and the -z-axis at angle $\phi$. This reflection is affected by the polarization of the incoming primary beam in both the horizontal (x) and vertical (y) directions.

for observed partial intensity $i$ of miller index $\boldsymbol{h}$.

## Scale factor, $G_0$ and $B$.
The derivatives of function $g$ with respect to $G_0$:

$$\frac{\partial g}{\partial G_0} = -\frac{e^{-2B(\sin\theta/\lambda)^2}\boldsymbol{Eoc}l}{\sigma V_c}. \tag{24}$$

The derivatives of function $g$ with respect to $B$:

$$\frac{\partial g}{\partial B} = -\frac{G_0\boldsymbol{Eoc}l}{\sigma V_c}\left[2\left(\frac{\sin\theta}{\lambda}\right)^2 e^{-2B(\sin\theta/\lambda)^2}\right]. \tag{25}$$

## Crystal rotation angles ($\theta_x$, $\theta_y$, $\theta_z$).
Although three rotation angles $\theta_x$, $\theta_y$, $\theta_z$ can be refined, a rotation around the beam direction (z-axis) has no component on the reciprocal-lattice offset ($r_h$) from the Ewald sphere—therefore, the derivative with respect to $\theta_z$ is 0. The partial derivatives with respect to the remaining parameters can be derived in a similar way—here, only the derivatives with respect to are $\theta_y$ given.

$$\frac{\partial g}{\partial\theta_y} = \frac{\partial g}{\partial\boldsymbol{Eoc}}\frac{\partial\boldsymbol{Eoc}}{\partial r_h}\frac{\partial r_h}{\partial\boldsymbol{x}}\frac{\partial\boldsymbol{x}}{\partial R}\frac{\partial R}{\partial\theta_y}, \tag{26}$$

where

$$\frac{\partial g}{\partial\boldsymbol{Eoc}} = \frac{-G_0 e^{-2B(\sin\theta/\lambda)^2}I}{\sigma V_c},$$

$$\frac{\partial\boldsymbol{Eoc}}{\partial r_h} = \frac{-4r_h r_s^2}{\left(2r_h^2 + r_s^2\right)^2},$$

$$\frac{\partial r_h}{\partial\boldsymbol{x}} = \frac{S}{|S|},$$

$$\frac{\partial\boldsymbol{x}}{\partial R} = \frac{\partial R}{\partial\theta_y}\boldsymbol{Oh},$$

and $R$ is the rotation matrix of the still image. The derivatives of the $g$ function (*Equation 24*) with respect to $\theta_x$ and the unit-cell parameters can be calculated by substituting the last partial derivatives of $R$ with the appropriate ones.

## Unit-cell parameters ($a$, $b$, $c$, $\alpha$, $\beta$, $\gamma$)
For unit-cell parameters, constraints imposed by crystallographic space groups are applied during the refinement—e.g., tetragonal systems only have two free parameters ($a$ and $c$) since $a = b$ and $\alpha = \beta = \gamma = 90$. Other restraint conditions, such as allowable refinement limits of the unit-cell dimensions, can also be applied as a 'penalty terms' in the least-squares refinement. The partial derivatives with respect to each unit-cell parameter in reciprocal units (here, $a^*$ is given and $a^* = 1/a$):

$$\frac{\partial g}{\partial a^*} = \frac{\partial g}{\partial\boldsymbol{Eoc}}\frac{\partial\boldsymbol{Eoc}}{\partial r_h}\frac{\partial r_h}{\partial\boldsymbol{x}}\frac{\partial\boldsymbol{x}}{\partial O}\frac{\partial O}{\partial a^*}, \tag{27}$$

where $\frac{\partial g}{\partial\boldsymbol{Eoc}}$, $\frac{\partial\boldsymbol{Eoc}}{\partial r_h}$, and $\frac{\partial r_h}{\partial\boldsymbol{x}}$ are as derived in (2) and

$$\frac{\partial\boldsymbol{x}}{\partial O} = R\frac{\partial O}{\partial a^*}h.$$

## Reflection radius, $r_s$

The reflection radius that accommodates effects of crystal mosaicity and spectral dispersion, described by the four parameters, $\gamma_0$, $\gamma_y$, $\gamma_x$, and $\gamma_e$, has following derivatives:

For $\gamma_y$,

$$\frac{\partial g}{\partial \gamma_y} = \frac{\partial g}{\partial Eoc} \frac{\partial EocV_c^{-1}}{\partial r_s} \frac{\partial r_s}{\partial \gamma_y}, \tag{28}$$

where $\frac{\partial g}{\partial Eoc}$ is derived in (*Equation 27*) and

$$\frac{\partial EocV_c^{-1}}{\partial r_s} = \frac{-3(r_s^2 + 2r_h^2)}{4(r_s^2 + 2r_h^2)^2},$$

$$\frac{\partial r_s}{\partial \gamma_y} = \frac{\gamma_y \cos^2 \alpha}{\left[ \left( \gamma_y \cos \alpha \right)^2 + (\gamma_x \sin \alpha)^2 \right]^{1/2}}.$$

For $\gamma_x$ and $\gamma_e$, the $\frac{\partial g}{\partial p}$ and $\frac{\partial p}{\partial r_s}$ are the same as derived for $\gamma_y$ and

$$\frac{\partial r_s}{\partial \gamma_x} = \frac{\gamma_x \sin^2 \alpha}{\left[ \left( \gamma_y \cos \alpha \right)^2 + (\gamma_x \sin \alpha)^2 \right]^{1/2}},$$

$$\frac{\partial r_s}{\partial \gamma_0} = 1,$$

$$\frac{\partial r_s}{\partial \gamma_e} = \tan \theta.$$

## Polarization correction

The XFEL beam is nearly 100% polarized in the horizontal direction. The optics at both the LCLS XPP and CXI stations do not introduce additional polarization. To account for the polarization of the primary beam, for a given reflection, we consider the angle $\phi$ between the sample reflection plane formed by the **h** vector and the -z-axis, and the laboratory horizontal (*Figure 14*).

As described in *Kahn et al. (1982)*, the beam $I_0$ incident on the sample crystal can be described in terms of two components, one parallel ($\sigma$) and the other perpendicular ($\pi$) to the plane of reflection:

$$I_0 = I_\sigma + I_\pi. \tag{29}$$

Each of these components is affected by the polarization of the primary beam in both the horizontal (x) and vertical (y) directions. Using $f_x$ and $f_y$ as the fractions horizontal and vertical in the laboratory frame ($f_x + f_y = 1$),

$$I_\sigma = \left( f_x \cos^2 \phi + f_y \sin^2 \phi \right) I_0, \tag{30}$$

and

$$I_\pi = \left( f_x \sin^2 \phi + f_y \cos^2 \phi \right) I_0, \tag{31}$$

where $f_x$ and $f_y$ are the polarization fractions in the x and y directions.

After reflection, only $I_\sigma$ is attenuated:

$$I' = I'_\pi + I'_\sigma = |F|^2 \left( I_\pi + I_\sigma \cos^2 2\theta \right). \tag{32}$$

By substituting $I_\sigma$ and $I_\pi$ from *Equations 30* and *31* in *Equation 32*, we arrive at

$$I' = |F|^2 \left[ f_x \left( \sin^2 \phi + \cos^2 \phi \cos^2 2\theta \right) + f_y \left( \cos^2 \phi + \sin^2 \phi \cos^2 2\theta \right) \right] I_0, \tag{33}$$

where the bracketed expression is P (*Kahn et al., 1982*).

## Molecular replacement and atomic model refinement protocol

To ensure atomic model refinements against the various diffraction data sets were as comparable as possible, we used a standard semi-automated solution and refinement protocol. First, we performed molecular replacement phasing with known structures as search models (PDB ID 3U3E for myoglobin, 3C8Y for hydrogenase, and 2TLI for thermolysin) with all heteroatoms, water molecules, and ligands removed. Molecular replacement was carried out with Phaser (*McCoy et al., 2007*) using default settings, with r.m.s.d. set to 0.8. The resulting solutions were then refined using *phenix.refine* (*Afonine et al., 2012*) in two cycles. In the first cycle, we carried out rigid body refinement, positional (xyz) refinement with automatic correction of Asn, Gln and His sidechain orientations, and atomic displacement parameter (ADP) refinement. We then used the difference density maps for missing ligands and heteroatoms obtained from this cycle to calculate real-space correlation coefficients using *phenix.get_cc_mtz_pdb* from the *PHENIX* software suite (*Adams et al., 2010*) for myoglobin and thermolysin and the program 'Map Correlation' from the CCP4 software (*Winn et al., 2011*) for hydrogenase. These omit difference density maps are shown in *Figures 6*, *7*, *8*, *10*. In the second cycle, all ligands and heteroatoms were placed in the difference density maps and combined with the refined structure from the first cycle using Coot (*Emsley et al., 2010*). The second cycle employed positional and ADP refinement with target weights optimization and water update was carried out with these complete models. The structures were validated by MolProbity (*Chen et al., 2010*). Final refinement statistics (*Tables 2, 3, 4*) were analyzed with *phenix.polygon* (*Urzhumtseva et al., 2009*) and found to be within acceptable range for other structures at similar resolutions. For the thermolysin structure obtained from anomalous diffraction data (processed keeping Friedel pairs separate), only one cycle of atomic model refinement was carried out. All figures were made in PyMOL (The PyMOL Molecular Graphics System, Version 1.5.0.4 Schrödinger, LLC.).

## Computer program

The computer program, *prime*, is implemented as a part of the *cctbx* computational crystallography toolbox (*Grosse-Kunstleve et al., 2002*). Download and installation instructions are available on the *cctbx* website (http://cctbx.sourceforge.net).

## Note added at proof

Subsequent to acceptance of this article, a paper was published by *Ginn et al. (2015)* describing an alternative method for orientation refinement as compared to the method of *Sauter et al. (2014)*, and partiality estimation for each individual image, but without post-refinement.

## Acknowledgements

We thank Henrik Lemke, Sebastien Boutet, and Ralf Grosse-Kunstleve for discussions. We thank S Michael Soltis, Aina E Cohen, Ana González, Yingssu Tsai, Winnie Brehmer, Laura Aguila, Jinhu Song, Scott McPhillips, and Henrik Lemke for providing the myoglobin XFEL diffraction data set. We thank John W Peters, Stephen Keable, Oleg A Zadvornyy, Aina E Cohen, S Michael Soltis, Jinhu Song, Scott McPhillips, Clyde Smith, and Henrik Lemke for providing the CpI hydrogenase XFEL diffraction data set. Portions of this research were carried out at the Linac Coherent Light Source (LCLS) at the SLAC National Accelerator Laboratory. LCLS is an Office of Science User Facility operated for the U.S. Department of Energy Office of Science by Stanford University. ASB and NKS were supported by National Institutes of Health grants GM095887 and GM102520 and Director, Office of Science, Department of Energy under contract DE-AC02-05CH11231. WIW was supported in part by National Institutes of Health grant P41 GM103393. This work is supported by a HHMI Collaborative Innovation Award (HCIA) to ATB and WIW.

## Additional information

### Competing interests

ATB: Reviewing editor, *eLife.* The other authors declare that no competing interests exist.

## Funding

| Funder | Grant reference | Author |
|---|---|---|
| National Institute of General Medical Sciences (NIGMS) | GM103393 | William I Weis |
| National Institute of General Medical Sciences (NIGMS) | GM095887 | Aaron S Brewster, Nicholas K Sauter |
| Howard Hughes Medical Institute (HHMI) | Collaborative Innovation Award | Axel T Brunger, William I Weis |
| U.S. Department of Energy (Department of Energy) | DE-AC02-05CH11231 | Aaron S Brewster, Nicholas K Sauter |
| National Institute of General Medical Sciences (NIGMS) | GM102520 | Aaron S Brewster, Nicholas K Sauter |

The funders had no role in study design, data collection and interpretation, or the decision to submit the work for publication.

## Author contributions

MU, OBZ, AYL, ATB, WIW, Conception and design, Analysis and interpretation of data, Drafting or revising the article; JH, ASB, NKS, Conception and design, Analysis and interpretation of data

### Author ORCIDs

Johan Hattne, http://orcid.org/0000-0002-8936-0912
Axel T Brunger, http://orcid.org/0000-0001-5121-2036
William I Weis, http://orcid.org/0000-0002-5583-6150

# Additional files

## Major dataset

The following previously published dataset was used:

| Author(s) | Year | Dataset title | Dataset ID and/or URL | Database, license, and accessibility information |
|---|---|---|---|---|
| Hattne J, et al., | 2014 | Accurate macromolecular structures using minimal measurements from X-ray free electron lasers. Nat Methods 11:545-548 | http://cxidb.org/id-23.html | Publicly available at the CXIDB Coherent X-ray Imaging Data Bank ID-23. |

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
