## [Decision Letter]

Thank you for sending your work entitled “Enabling X-ray Free Electron Laser Crystallography for Challenging Biological Systems from a Limited Number of Crystals” for consideration at *eLife*. Your article has been favorably evaluated by John Kuriyan (Senior editor) and three reviewers, one of whom, Stephen Harrison, is a member of our Board of Reviewing Editors.

The Reviewing editor and the other reviewers discussed their comments before we reached this decision, and the Reviewing editor has assembled the following comments to help you prepare a revised submission.

The manuscript represents a substantial advance in the processing of XFEL diffraction data, through the introduction of postrefinement methods. Although it provides no direct comparison with other procedures (from Kabsch or White), data for the three test cases considered show considerably improved statistics. Moreover, the total number of frames required to compile a data set is very much smaller than with the so-called “Monte Carlo” method. The manuscript is well written and thorough (especially in the provision of equations in the Methods section). A possible, but forgivable, omission was the inclusion of a more challenging test case that could demonstrate a genuine increase in the effective resolution of a dataset from the new procedures.

The reviewers had the following modest concerns, questions and requests:

A. Theory:1) Sphere model for *Eoc*_*h*_. The basic idea of postrefinement is that the ratio of the intensity of a partially recorded reflection to its value in a properly corrected and scaled reference data set is a very sensitive measure of the orientation and unit-cell parameters of the crystal and of diffracting-range parameters such as mosaic spread, energy dispersion, and range of unit-cell parameters within the diffracting volume. [Disp-formula equ1] and [Disp-formula equ2] summarize the application to XFEL stills, with *Eoc*_*h*_ as the critical function requiring definition and evaluation. The authors choose a sphere model for *Eoc*, which is reasonable for cases in which the diffracting range is dominated by energy dispersion and unit-cell variation. For a mosaic crystal with no variation in unit cell across the diffracting volume and mosaicity higher than the energy dispersion, the reciprocal-space shape of the diffraction spot will not be spherical; it will intersect the Ewald sphere as an arc, since the Bragg angle and hence the distance of any component of the spot from the origin of reciprocal space will be (ex hypothesi) invariant, while the range of azimuthal angles of the spot on the detector will depend on the mosaic spread (assumed to be non-zero). With cryopreserved crystals, the assumption that a combination of unit-cell variation and energy dispersion dominates is almost certainly a good one, but it may not hold for tiny crystals at ambient temperature in an injected beam. Anisotropy of some of the parameters may also make other shapes a better fit. The approach in the paper is, of course, generalizable to other shapes (with much “hairier” expressions for *Eoc*_*h*_ and its derivatives). In any case, the authors should discuss the assumptions that go into the sphere approximation.

2) Lorentz factor. A clear discussion of Lorentz factor is important, to give the paper full archival value as a complete treatment of the intensity correction problem. Formally, there is no Lorentz factor for a still. This statement is easy to prove using the “sinc” formula given in [Disp-formula equ1] of the cited article by [19]. If two different relps lie precisely on the Ewald sphere, then the value of the sinc function is simply equal to the square of the number of unit cells, regardless of resolution or any other geometric factor. All that remains is the polarization (which is not a Lorentz factor) and the incident intensity, which is the same for every spot. The only terms that remain hkl-dependent are the structure factor F, and the solid angle subtended by a pixel. The latter has some semblance to a Lorentz factor, but disappears upon pixel integration if the detector is corrected to be spherical. The spreading out of the spot due to mosaic spread and spectral dispersion in reciprocal space could be considered a Lorentz factor, but in the context of the present work, this should be part of the “partiality”.

B. Questions:

1) In the test cases, the data quality for the subset of images (e.g. 2,000 for thermolysin) is clearly lower than using the entire dataset. Is there any indication of convergence when considering data quality metrics vs the number of images included, or does inclusion of all images always give the best data?

2) The Discussion section is relatively brief. Even with the improved processing, the data quality falls significantly short of what would be expected for conventional SR rotation data collection. Does the analysis provide any pointers to the remaining major sources of error?

3) Figure 6 (myoglobin data): For the high resolution terms, post-refinement appears to make the data worse as judged by the *R* and *R*_*free*_ metrics. Why?

4) There are many fewer spots per image for thermolysin than for the other two datasets. What is the definition of a “spot” in this context?

5) It is not clear if a separate resolution limit is applied to each image during the final merging step. Can this be clarified?

6) Figure 9: What is the second peak that is clearly visible when all images are used? Perhaps it would be useful to quote the largest “noise” peak as well as that for the Zn.

7) Table 3: The hydrogenase data were collected with a seeded beam, and yet the term representing the energy dispersion γe is larger than that for thermolysin and almost as large as for the myoglobin data. Why?

C. Request:

The paper should have a complete list of all the parameters and symbols in the equations and their definitions (as Acta Cryst may still do and certainly used to do). Many of the parameters (such as theta(x) and theta(y)) were defined only in the figures, and it might indeed clutter the text to define each of them immediately after their first appearance in [Disp-formula equ1].

---

## [Author Response]

*A. Theory*:

*1) Sphere model for* Eoc_h_*. The basic idea of postrefinement is that the ratio of the intensity of a partially recorded reflection to its value in a properly corrected and scaled reference data set is a very sensitive measure of the orientation and unit-cell parameters of the crystal and of diffracting-range parameters such as mosaic spread, energy dispersion, and range of unit-cell parameters within the diffracting volume.*
[Disp-formula equ1] and [Disp-formula equ2]
*summarize the application to XFEL stills, with* Eoc_h_
*as the critical function requiring definition and evaluation. The authors choose a sphere model for* Eoc*, which is reasonable for cases in which the diffracting range is dominated by energy dispersion and unit-cell variation. For a mosaic crystal with no variation in unit cell across the diffracting volume and mosaicity higher than the energy dispersion, the reciprocal-space shape of the diffraction spot will not be spherical; it will intersect the Ewald sphere as an arc, since the Bragg angle and hence the distance of any component of the spot from the origin of reciprocal space will be (ex hypothesi) invariant, while the range of azimuthal angles of the spot on the detector will depend on the mosaic spread (assumed to be non-zero). With cryopreserved crystals, the assumption that a combination of unit-cell variation and energy dispersion dominates is almost certainly a good one, but it may not hold for tiny crystals at ambient temperature in an injected beam. Anisotropy of some of the parameters may also make other shapes a better fit. The approach in the paper is, of course, generalizable to other shapes (with much* “*hairier*” *expressions for* Eoc_h_
*and its derivatives). In any case, the authors should discuss the assumptions that go into the sphere approximation*.

The spherical model used in this work is indeed a crude approximation of the diffraction spots. In the Discussion, we now describe the factors that contribute to spot shape and the consequent limitations of our model that will need to be addressed in the future.

*2) Lorentz factor. A clear discussion of Lorentz factor is important, to give the paper full archival value as a complete treatment of the intensity correction problem. Formally, there is no Lorentz factor for a still. This statement is easy to prove using the* “*sinc*” *formula given in*
[Disp-formula equ1]
*of the cited article by*
[19]*. If two different relps lie precisely on the Ewald sphere, then the value of the sinc function is simply equal to the square of the number of unit cells, regardless of resolution or any other geometric factor. All that remains is the polarization (which is not a Lorentz factor) and the incident intensity, which is the same for every spot. The only terms that remain hkl-dependent are the structure factor F, and the solid angle subtended by a pixel. The latter has some semblance to a Lorentz factor, but disappears upon pixel integration if the detector is corrected to be spherical. The spreading out of the spot due to mosaic spread and spectral dispersion in reciprocal space could be considered a Lorentz factor, but in the context of the present work, this should be part of the* “*partiality*”.

We agree that there is no Lorentz correction for a stationary crystal and monochromatic beam. We had tried to follow the discussion of [17] on this point, but now we explicitly note that there is no Lorentz correction for a still, and have removed the discussion of the Kabsch paper on this topic.

*B. Questions*:

*1) In the test cases, the data quality for the subset of images (e.g. 2,000 for thermolysin) is clearly lower than using the entire dataset. Is there any indication of convergence when considering data quality metrics vs the number of images included, or does inclusion of all images always give the best data*?

In the last section of Results we now describe a comparison of thermolysin diffraction data sets merged using 2,000-12,000 images. To avoid potential differences arising from different levels of completeness, which would confound this analysis, we truncated the diffraction data at 2.6 Å to insure that each of these sets was 100% complete. Comparison of *CC*_*1/2*_ values, electron density maps and model *R* values shows that there is little improvement beyond 8,000 images.

*2) The Discussion section is relatively brief. Even with the improved processing, the data quality falls significantly short of what would be expected for conventional SR rotation data collection. Does the analysis provide any pointers to the remaining major sources of error*?

This is an excellent question, but we do not feel that we can directly compare SR rotation and XFEL data at this point. The one comparison that we present (Figure 7) suggests that the SR data are at least somewhat better, but it is difficult to quantify. It is likely that rotation data would be better due to the ability to directly measure full reflections (at least by summation of partials) without modeling partiality, which is still a relatively crude process. However, we believe that a comparison between still diffraction data sets collected at SR and XFELs is needed to deconvolute the effect of rotation vs. other differences between SR and XFEL sources. This will be a subject of future investigation. We have added a brief discussion of these issues to the new Discussion.

*3)*
Figure 6
*(myoglobin data): For the high resolution terms, post-refinement appears to make the data worse as judged by the* R *and* R_free_
*metrics. Why*?

We assume that the question refers to the 100 image subset; the full post-refined 757 image set has lower or comparable *R* values in all bins. The 100-image set is the minimum number of images required for successful molecular replacement and an interpretable omit map (the heme group). While the *R* and *R*_*free*_ values of the 757-image set (97.7% completeness) improved in all resolution shells, they improved only to approximately 1.7 Å, where the completeness drops below 90%. We observed that completeness has an impact on post-refinement procedure and the post-refined data sets. We now note this effect in the last section of Results, “Effect of completeness”.

*4) There are many fewer spots per image for thermolysin than for the other two datasets. What is the definition of a* “*spot*” *in this context*?

We now describe the criteria for “spot” definition in the subsection headed “Preparation of the observed intensities”.

*5) It is not clear if a separate resolution limit is applied to each image during the final merging step. Can this be clarified*?

The cctbx.xfel program applies separate resolution cutoffs on each image. This is now noted in the Results section.

*6)*
Figure 9*: What is the second peak that is clearly visible when all images are used? Perhaps it would be useful to quote the largest* “*noise*” *peak as well as that for the Zn*.

We thank the reviewer for pointing out this feature, which turns out not to be noise. We suggest that the second anomalous peak may indicate a second zinc ion: indeed, a previous thermolysin structure (PDB ID: 1LND; [15]) has two zinc ions bound to the same active site, and their locations match with the anomalous peaks observed in the post-refined maps of Figure 9. We re-refined the thermolysin structure with two zinc ions (the refined *B*-factors for the first zinc ion with occupancy 1.0 and the second zinc ion with occupancy 0.5 are 24.4 and 30.9, respectively). Interestingly, adding the second zinc ion resulted in an improvement of difference density of the dipeptide near the zinc sites (see Table 4 for updated refinement statistics). Thus, in addition to the missing dipeptide in the original structure (PDB ID: 4OW3; [13]), this adds another feature that was not clearly visible before the post-refinement procedures.

*7)*
Table 3*: The hydrogenase data were collected with a seeded beam, and yet the term representing the energy dispersion γe is larger than that for thermolysin and almost as large as for the myoglobin data. Why*?

We mistakenly thought that these data had been measured with a seeded beam, as a single energy value was present in the header records of each frame. However, after conferring with the experimental team that collected the data (7), we discovered that in fact the data were collected with the usual SASE spectrum; there was a hardware problem that prevented recording the energy spectrum per frame. We have revised our manuscript accordingly.

*C. Request*:

*The paper should have a complete list of all the parameters and symbols in the equations and their definitions (as Acta Cryst may still do and certainly used to do). Many of the parameters (such as theta(x) and theta(y)) were defined only in the figures, and it might indeed clutter the text to define each of them immediately after their first appearance in*
[Disp-formula equ1].

We have added a full list of parameters and symbols with their definitions in the Notation section.